# Non-Linear Modeling of Detectability of Ship Wake Components in Dependency to Influencing Parameters Using Spaceborne X-Band SAR

**Björn Tings** [ID]

German Aerospace Center, Am Fallturm 9, 28359 Bremen, Germany; bjoern.tings@dlr.de

**Abstract:** The detection of the wakes of moving ships in Synthetic Aperture Radar (SAR) imagery requires the presence of wake signatures, which are sufficiently distinctive from the ocean background. Various wake components exist, which constitute the SAR signatures of ship wakes. For successful wake detection, the contrast between the detectable wake components and the background is crucial. The detectability of those wake components is affected by a number of parameters, which represent the image acquisition settings, environmental conditions or ship properties including voyage information. In this study the dependency of the detectability of individual wake components to these parameters is characterized. For each wake component a detectability model is built, which takes the influence of incidence angle, polarization, wind speed, wind direction, sea state (significant wave height, wavelength, wave direction), vessel's velocity, vessel's course over ground and vessel's length into account. The presented detectability models are based on regression or classification using Support Vector Machines and a dataset of manually labelled TerraSAR-X wake samples. The considered wake components are: near-hull turbulences, turbulent wakes, Kelvin wake arms, Kelvin wake's transverse waves, Kelvin wake's divergent waves, V-narrow wakes and ship-generated internal waves. The statements derived about wake component detectability are mainly in good agreement with statements from previous research, but also some new assumptions are provided. The most expressive influencing parameter is the movement velocity of the vessels, as all wake components are more detectable the faster vessels move.

**Keywords:** detectability model; machine learning; Synthetic Aperture Radar; wake detection

## 1. Introduction

Using Synthetic Aperture Radar (SAR) imagery, moving objects on the ocean surface can be detected directly or indirectly. Direct detection means searching for the object's signatures itself, e.g., [1]. Indirect detection means searching for wakes, which are caused by the object's movements on the ocean surface, e.g., [2]. As moving objects are normally ships, in this paper all kinds of wakes caused by moving objects are referred to by "ship wakes". The detection of ship wakes can be performed through a variety of methods and has been investigated for decades, e.g., [3]. Most state-of-the-art wake detection methods are based on analytic functions, often using a Radon Transform [4,5], but in recent years methods from the field of data science emerged, e.g., [6,7].

Nowadays, the availability and utilization of large amounts of sensor data is quite common. While some researchers have taken advantage of developments in data science for wake detection, research using data science techniques to analyze the general appearance of wakes in SAR data and the related wake detectability is sparse. The first studies about the detectability of ship wakes in relation to parameters influencing the detectability were based on simulations and theoretical considerations [8–10]. Those parameters, which describe environmental conditions, acquisition settings or ship properties, are in the following denoted "influencing parameters". Approaches applying data science for the analysis of detectability in relation to influencing parameters have only recently been published

in [11,12]. However, these two data science studies do not differentiate between the individual wake components recognizable in SAR data, while studies based on simulation and theoretical considerations do differentiate between turbulent wake, Kelvin wake arms, V-narrow wake, ship-generated internal wave, divergent waves and transverse waves [8]. Therefore, this paper describes how data science can be applied to test the statements derived from simulations and theory about the detectability of individual wake components in dependency to influencing parameters. Another improvement in comparison to [11,12] is the utilization of the wake's extent as indicator for detectability, instead of using binary flags. Additionally, is shown here that the applied method can be used:

1.    to derive new assumptions about influencing parameters not considered in the past
2.    to derive new assumptions about interdependent influences of the parameters.

The data used in this study was acquired by the TerraSAR-X (TS-X) spaceborne SAR mission. As TS-X has an X-Band SAR sensor mounted onboard, the focus of this study is on X-Band SAR data only. However, comparisons between C-Band and X-Band spaceborne SAR missions are provided in [11,13]. This means a generalization of statements to C-Band data is partially possible.

In the following two subsections the state-of-the-art research on wake components and their detectability for X-Band SAR is summarized.

### 1.1. Wake Components Detectable on SAR

The following subsections summarize the state-of-the-art of science on the topic of ship wakes in SAR images. In Figure 1 all wake components visible in SAR imagery are visualized. As in the literature no uniform naming convention for the individual wake components exists, the wake components in Figure 1 are denoted and colored as follows: the port side versions of Kelvin wake arm (red), V-narrow wake arm (light orange) and ship generated internal wave (light purple) are highlighted with brighter colors than their starboard counterparts, i.e., Kelvin wake arm (dark red), V-narrow wake arm (orange) and ship generated internal wave (purple). The Kelvin wake's transverse waves are visualized in cyan and the divergent waves in blue. In this study the near-hull turbulence (light green) i.e., near field of the wake (which is also known as wake generation region [14]) is used as separate wake component. The calmer ocean surface region in the centerline of the wake is in the following denoted turbulent wake (green). More details about the individual wake components are provided in the following subsections.

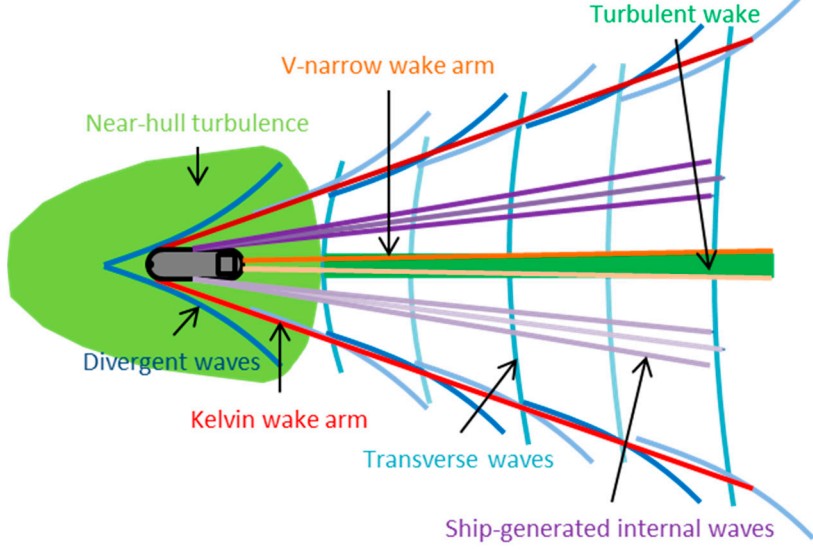

**Figure 1.** Schematic visualization of all ship wake components visible in Synthetic Aperture Radar (SAR) imagery.

### 1.1.1. Turbulent Wake

The most frequently occurring wake component is the turbulent wake. The ocean surface area aft of the ship is affected by the ship's transit. In remote sensing data the induced turbulent wake signature can reach up to tens of kilometers in length following the ship's path [14,15]. It consists of two parts. First, a region containing whitewater and rough ocean surface directly aft of the ship, which is a result of the turbulences created by the ship's propellers. Second, a calm ocean surface region attached to this region, which is a result of attenuation of short ambient ocean surface waves by ascending bubbles and surfactants.

Reed et al. [14] use the term "near field" to describe a region around moving ships, which produce higher radar backscattering compared to the unaffected backscatter of the surrounding ocean surface. The near field does not only contain the turbulences from the propeller wake aft the ship, but also the turbulences created at the ship's bow waves and hull drag to the front and at side of the ship. Instead of considering all kinds of whitewater and rough ocean surface directly attached to the ship's hull as a region, here those turbulences are considered as a single wake component and denoted near-hull turbulence (see Figure 2). It should explicitly be noted that near-hull turbulence also contains the smearing of the propeller wake's radar signatures, which occur along the azimuth direction between the wake vertex and the focused ship in case of Doppler azimuth displacement of the ship's signature [16–19].

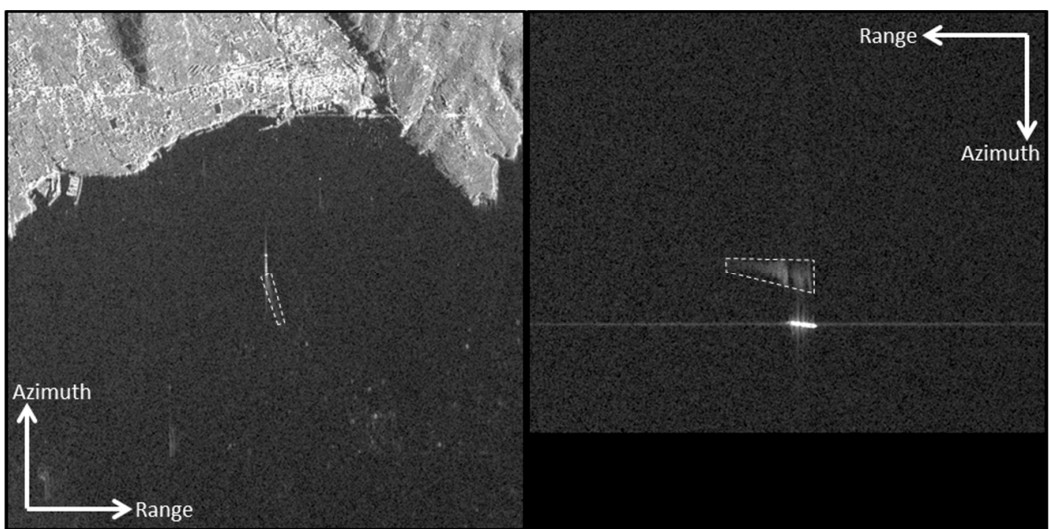

**Figure 2.** Examples for the wake component "near-hull turbulence" with ship movement parallel to azimuth direction (**left**) and parallel to range direction (**right**), approximately, both acquired with TS-X using Stripmap-mode, HH-polarization and pixel spacing of 3 m.

In the following, turbulent wake will only refer to the calm water region with low backscatter (see Figure 3), which appears darker in SAR images compared to the unaffected backscatter of the surrounding ocean surface [14,15,20].

### 1.1.2. Kelvin Wake

The Kelvin wake is a complex pattern on the ocean surface, which originates from an interaction between two wake components: the transverse waves and the divergent waves [21,22]. Constructive interference of both wave systems results in so called cusp waves, which are located on two lines on a V-shaped envelope originating symmetrically from the ship hull [8,9]. Often, they even appear as two bright lines, instead of as multiple single waves. However, cusp waves are single waves with high wave amplitude. Therefore, they often result in wave breaking. In SAR imagery these cusp waves create regions of higher backscatter compared to the surrounding ocean surface and are therefore frequently

detectable. Signatures of transverse and divergent waves occur less often in SAR imagery than the cusp waves they create. While Kelvin wake refers to the full V-shaped pattern consisting of transverse waves, divergent waves and cusp waves, the two regions with high backscattering ocean surface located on the two V-lines are denoted "Kelvin wake arms" [9].

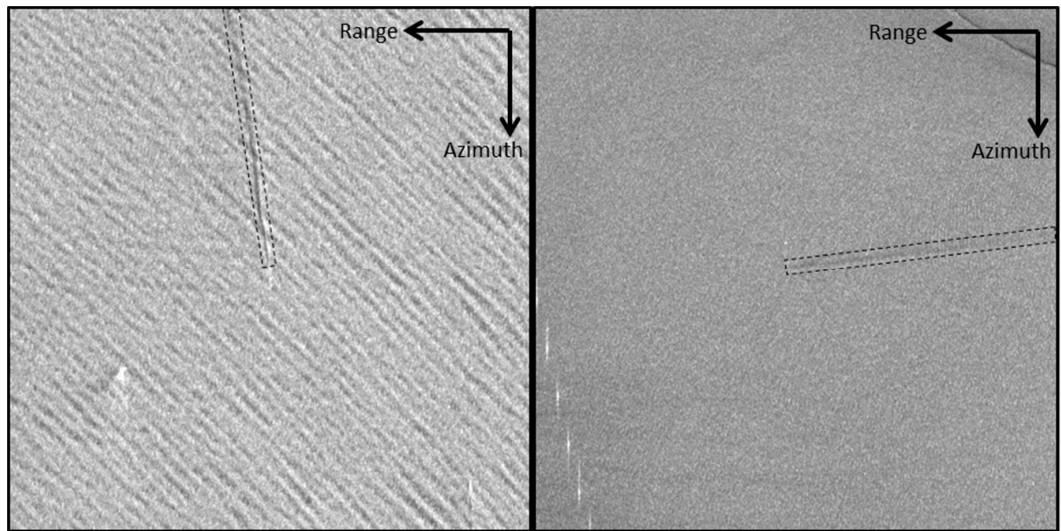

**Figure 3.** Examples for the wake component "turbulent wake" with ship movement parallel to azimuth direction (**left**) and parallel to range direction (**right**), approximately, both acquired with TS-X using Stripmap-mode, HH-polarization and pixel spacing of 3.25 m/3 m (**left/right**).

As the Kelvin wake arms do not always appear symmetrically in SAR images, the starboard and port Kelvin wake arms are investigated separately. The Kelvin wake mostly originates from the ship's bow, but sometimes a second starboard and/or port Kelvin arm is detectable, which originates from the ship's stern. In theory the angle between the ship's course over ground (CoG) and the Kelvin wake arms corresponds to 19.47°, but with certain ship properties the angle can be smaller [21]. Figure 4 presents examples of Kelvin wakes with typical half angles of the Kelvin wake arms. In Figures 5 and 6 examples of divergent waves and transverse waves are presented, respectively.

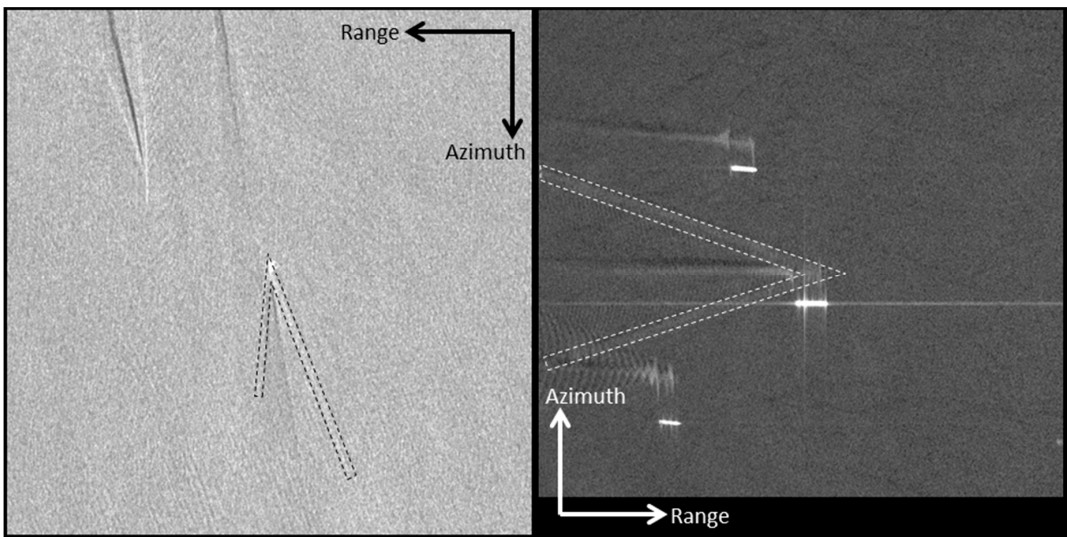

**Figure 4.** Examples for the wake component "Kelvin wake arm" with ship movement parallel to azimuth direction (**left**) and parallel to range direction (**right**), approximately, both acquired with TS-X using Stripmap-mode, HH-polarization and pixel spacing of 3.25 m/2.75 m (**left/right**).

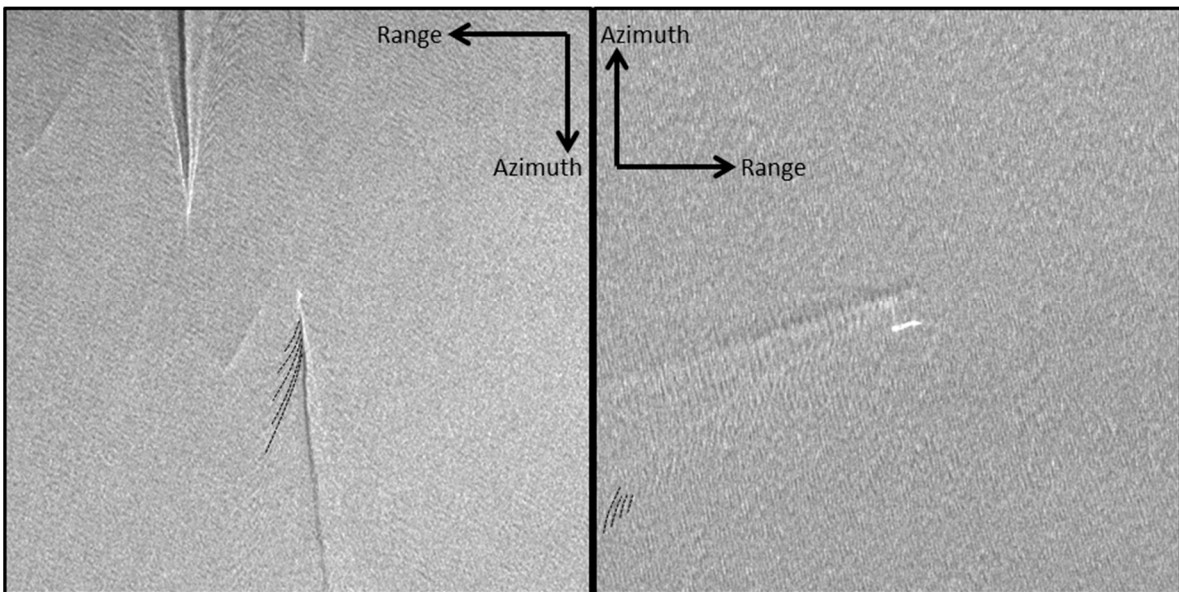

**Figure 5.** Examples for the wake component "divergent waves" with ship movement parallel to azimuth direction (**left**) and parallel to range direction (**right**), approximately, both acquired with TS-X using Stripmap-mode, HH-polarization and pixel spacing of 3.25 m/3 m (**left/right**).

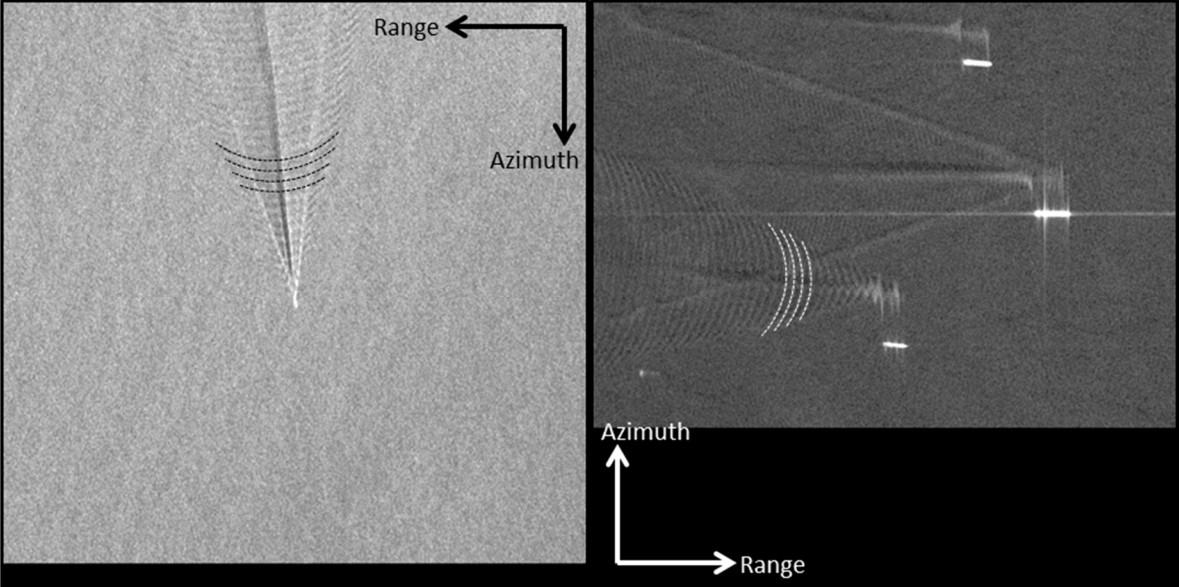

**Figure 6.** Examples for the wake component "transverse waves" with ship movement parallel to azimuth direction (**left**) and parallel to range direction (**right**), approximately, both acquired with TS-X using Stripmap-mode, HH-polarization and pixel spacing of 3.25 m/2.75 m (**left/right**).

### 1.1.3. V-Narrow Wake

In 1988, ref. [8] provided an explanation for the other bright V-shaped pattern with narrow angle to both sides of the turbulent wake's calm water region, the so called V-narrow wakes, which is now widely supported by other researchers and their measurements [14]. Ref. [8] assumes that ship-generated surface waves propagate circularly from their point of origin into all directions. The ship's movement would then generate multiple wave origin points, while their repetition frequency can satisfy the first- or second-order Bragg criterion with the parameters of the observing radar. When the Bragg criteria are met, Bragg scattering occurs in the form of bright lines with narrow angle aft of the ship. This

explanation for the presence of V-narrow wakes in SAR imagery is also supported by [16]. However, a series of explanations for this effect exists. In this study it is the one adopted, as it best explains the obtained results. In [17], two further explanations are mentioned, which are not considered here.

Ref. [8] also assumes that V-narrow wakes are not detectable on X-Band SAR, as their Bragg waves would be too short for sufficient lifetime or they would be consumed by the turbulent wake. This statement is also supported in other publications [18]. Indeed, most of the research from around that time only investigated the V-narrow wakes on the basis of L-Band data [16,18]. However, for this study a manual inspection of several thousands of wake samples was executed and in TS-X images V-narrow wakes were found frequently.

Figure 7 presents how V-narrow wakes appear on TS-X's X-Band SAR. The V-narrow wakes from ships with movement parallel to azimuth show the described characteristic pattern similar to the L-Band V-narrow wakes. In contrast, the V-narrow wakes from ships with movement parallel to range differ from their L-Band counterparts, as mostly only one V-narrow wake arm is detectable due to bright scattering. Additionally, the V-narrow wake arms based on movement parallel to range appear rather blurry and are not as distinct compared to the V-narrow wake arms based on movement parallel to azimuth. Why these patterns are still considered as V-narrow wake, instead of part of the turbulent wake, is discussed in Section 4.9.

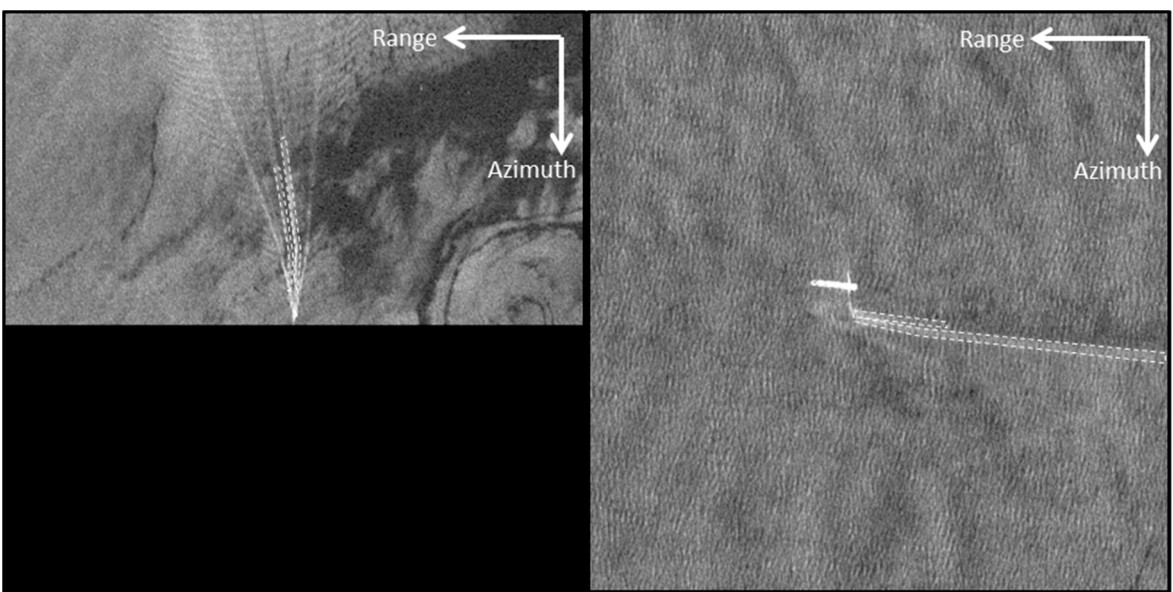

**Figure 7.** Examples for the wake component "V-narrow wake" with ship movement parallel to azimuth direction (**left**) and parallel to range direction (**right**), approximately. The port V-narrow wake arm of the left vessel is much shorter than its starboard arm, both acquired with TS-X using Stripmap-mode, HH-polarization and pixel spacing of 3.25 m/3 m (**left/right**).

### 1.1.4. Ship-Generated Internal Waves

Ship-generated internal waves appear as repeated V-shaped patterns alternating between bright and dark regions. Figure 8 presents how this wake component is imaged by TS-X. Only for the sake of completeness is the detectability of ship-generated internal waves considered here. It was known beforehand that this wake component only appears rarely, because its appearance requires strongly stratified water conditions near the surface [18,19,23,24]. As the water stratification is not considered as an influencing parameter in this study and in the study areas water stratification does not appear regularly, the main influencing factor for analysis of ship-generated internal waves is missing.

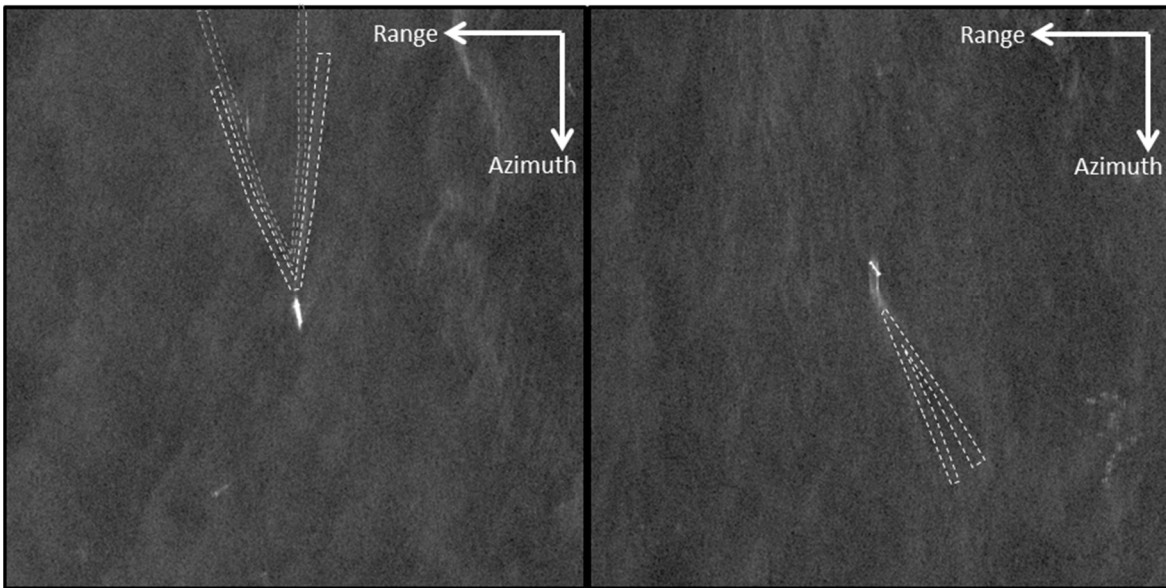

**Figure 8.** Examples for the wake component "ship-generated internal wave" with ship movement in approximately azimuth direction (**left**) and approximately 45° angle between range and azimuth direction (**right**), both acquired with TS-X using Stripmap-mode, HH-polarization and pixel spacing of 3 m. No ship-generated internal wave samples of ships moving parallel to range were found.

### 1.2. Statements on Detectability of Wake Components

Many publications about the dependency of the detectability of ship wake components on influencing parameters derive their results from simulations or theoretical considerations. Most of those results are in good agreement with each other. An overview of basic assumptions and statements developed before are provided in the following list. Details can be found in the Section 1.2, where the state-of-the-art is summarized. In the conclusion in Section 5 the following overview is updated according to the results obtained in this study.

- Turbulent wakes (including near-hull turbulences):
  - ○ are better detectable, when the vessels move parallel to azimuth direction
  - ○ are better detectable, when the incidence angles are smaller
  - ○ are better detectable, when the wind speeds are lower
  - ○ are better detectable, when the sea state's significant wave heights are lower
  - ○ are better detectable, when the wind direction is parallel to the ship's CoG
- Kelvin wake arms:
  - ○ are better detectable, when the vessels move faster
  - ○ are better detectable, when the vessels are larger
  - ○ are better detectable, when the vessels move parallel to azimuth direction
  - ○ are better detectable, when the incidence angles are smaller
  - ○ are better detectable, when the wind speeds are lower
  - ○ are better detectable, when the sea state's significant wave heights are lower
  - ○ are better detectable, when the wind direction is perpendicular to the ship's CoG
- Kelvin wake's divergent waves:
  - ○ are better detectable, when the vessels move faster
  - ○ are better detectable, when the vessels are smaller
- Kelvin wake's tranverse waves:
  - ○ are better detectable, when the vessels move slower
  - ○ are better detectable, when the vessels are larger
  - ○ are better detectable, when the vessels move parallel to azimuth direction

- V-narrow wake arms:
  - are better detectable, when the vessels move faster
  - are better detectable, when the vessels move parallel to azimuth direction
  - are better detectable, when the incidence angles are smaller
  - are better detectable, when the wind speeds are lower
  - are better detectable, when the sea state's significant wave heights are lower
- Ship-generated internal waves:
  - are mainly influenced by the water stratification, which is not considered as an influencing parameter in this study
  - are better detectable, when the vessels are larger
  - are better detectable, when the wind speeds are lower

### 1.2.1. Detectability of Turbulent Wakes

The detectability of calm water regions visible due to smoothed ocean surfaces decreases with increasing incidence angles [25]. Although in [25] this was stated for surface slicks, it also holds for the detectability of turbulent wakes, as these also produce a smoothed ocean surface.

The ocean surface has to be affected by moderate wind conditions [8], e.g., according to [15] above 2.5 m/s, for a turbulent wake's SAR signature to be distinctive. Otherwise, no ambient capillary waves are present on the ocean surface, whose production by striking winds is attenuated in the calm water region of the turbulent wake making it detectable. In [12] it is stated that higher wind speeds decrease the probability of detecting turbulent wakes, but the study is based on a similar data-driven approach than the one presented here. No other statements relating a higher or lower detectability of turbulent wakes with respect to higher or lower wind speed conditions were found in the literature. However, it is possible to substitute the limited statements about the dependency between wind speed and the detectability of turbulent wakes by studies about the detectability of surface films, as these are also detectable on SAR due to the attenuated production of ambient surface waves. According to [26,27], higher wind speeds decrease the dampening behavior of surface films. According to [18], at least for large vessels a wind direction parallel to the ship's course over ground (CoG) increases the turbulent wake's detectability.

As wind speed and sea state conditions are strongly correlated [28], a similar relationship between wave height and the detectability of turbulent wakes is present. According to [10,18,29] turbulent wakes are better detectable in low than in moderate sea state conditions.

The relative angle between the radar beam looking direction and ship's CoG has low influence on the detectability of turbulent wakes, but Lyden et al. [8] assumed that a relative looking direction parallel to the ship's CoG is worse for turbulent wake's detection than a perpendicular one.

### 1.2.2. Detectability of Kelvin Wakes

The incidence angle is of comparable importance with respect to detectability for both turbulent wakes and Kelvin wake arms. Thus, the detectability of Kelvin wake arms also decreases with increasing incidence angle [9].

Similar to turbulent wakes, Kelvin wakes also require a minimum wind speed during the SAR acquisitions, otherwise the ocean surface provides insufficient backscatter [8]. On the other hand, a rougher sea surface induced by higher wind speeds in the surrounding of Kelvin wake arms decreases the contrast between the cusp waves on the arms and the background. Therefore, Kelvin wake arms are better detectable under low wind speed conditions [9], except when no wind is present. Hennings et al. [9] further found the wind direction relative to the ship's CoG of less importance than the wind speed.

With high correlation between wind speed and sea state conditions, the detectability of Kelvin wake arms also decreases with increasing sea state heights [10,18]. Tunaley et al. [18] state that the velocity bunching effect [30] would be responsible. In [9] it is

additionally shown that the wind direction relative to the cusp waves increases the Kelvin arm's backscatter, when the direction is perpendicular.

The cusp waves on single Kelvin arms are most distinctive from the ocean background when their traveling direction is parallel to the range direction, which means Kelvin wake arm and range direction are perpendicular [31]. According to [9] cusp waves show increased backscatter, when traveling towards the sensor against the range direction, and decreased backscatter, when traveling away from the sensor in range direction. The Kelvin envelope with both Kelvin wake arms is best detectable, when the ship's CoG is parallel to the azimuth direction [8]. As the angle between CoG and the Kelvin wake arms mostly corresponds to roughly $\pm 20°$ [21,22], the single Kelvin wake arms then would have an angle relative to the range direction of 70° and 110°, respectively. Lyden et al. [8] also postulate that the Kelvin wake's transverse waves are best detectable with ship movement parallel to azimuth.

The propagation of Kelvin wakes can be described using the wake's Froude number [21,22]. The Froude number is a non-dimensional measure for the wave drag behind the ship. It is calculated using length and velocity of a ship by:

$$Fr = V / \sqrt{gL} \tag{1}$$

where $V$ is the vessel's velocity, $L$ the vessel's length and $g$ the gravitational acceleration. As the length only contributes to the Froude number under the square root, the vessel velocity has a stronger impact on the Froude number than the ship length. According to [21,22] the amplitude of the transverse waves decreases for larger Froude numbers and the V-shaped wave pattern becomes narrower due to the compression of the divergent waves. This means, due to the resulting higher radar backscatter from higher wave amplitude of the divergent waves, which contribute strongly to imaging of the Kelvin wake arms, that Kelvin wake arms of faster vessels would be better detectable. However, for vessels not exceeding their hull velocity, the impact of the vessel length on the amplitude of the divergent waves is important, as the constructive interference between transverse and divergent waves builds higher cusp waves [21,22], when both wave types are high. This means that a longer vessel's length would also result in better Kelvin wake arm detectability.

### 1.2.3. Detectability of V-Narrow Wakes

Lyden at al. [8] found more samples of V-narrow wakes under low incidence angle conditions and therefore assume that the V-narrow wake's detectability decreases for increasing incidence angles.

Furthermore, in [8] it is stated that the detectability is strongly dependent on wind speed. In L-Band data, V-narrow wakes were rarely found with wind speeds above 3 m/s. This influence is explained by increased nonlinear ambient waves coming with higher winds and resulting in stronger decay of the V-narrow wake arms. Therefore, calm sea state conditions would also imply better detectability.

Many researchers identified the angle between the ship's track and the radar looking direction as crucial for the detectability of ship wakes. For example, in [8,14,32] it is stated that the V-narrow wake arms are more often encountered with CoG parallel to azimuth direction than CoG parallel to range direction.

As the decay rate of the V-narrow wake arms is also time-dependent, a faster vessel should create longer V-narrow wake arms. Ref. [8] also found evidence for this influence.

The statements in this subsection are supported by an all-embracing analysis conducted by Zilman et al. in [17]. Unfortunately, no definite published statements can be found about the influence of the ship length on the detectability of V-narrow wakes.

### 1.2.4. Detectability of Ship-Generated Internal Waves

The detectability of ship-generated internal waves mainly depends on the stratification of the water body near the surface [19]. A minimum sea surface roughness, e.g., induced by small wind speeds, is required, but above this threshold the detectability decreases with

increasing wind speeds [23]. With higher vessel velocity the angle of the interval waves gets smaller, while the detectability is hardy affected [23]. According to [33] a larger ship size also influences the detectability by increasing the internal wave's amplitude.

## 2. Materials and Methods

Figure 9 describes the procedure from data generation (Figure 9A), influencing parameters extraction (Figure 9B) up to modelling and visualization (Figure 9C). The individual processing steps are explained in detail in the following subsections.

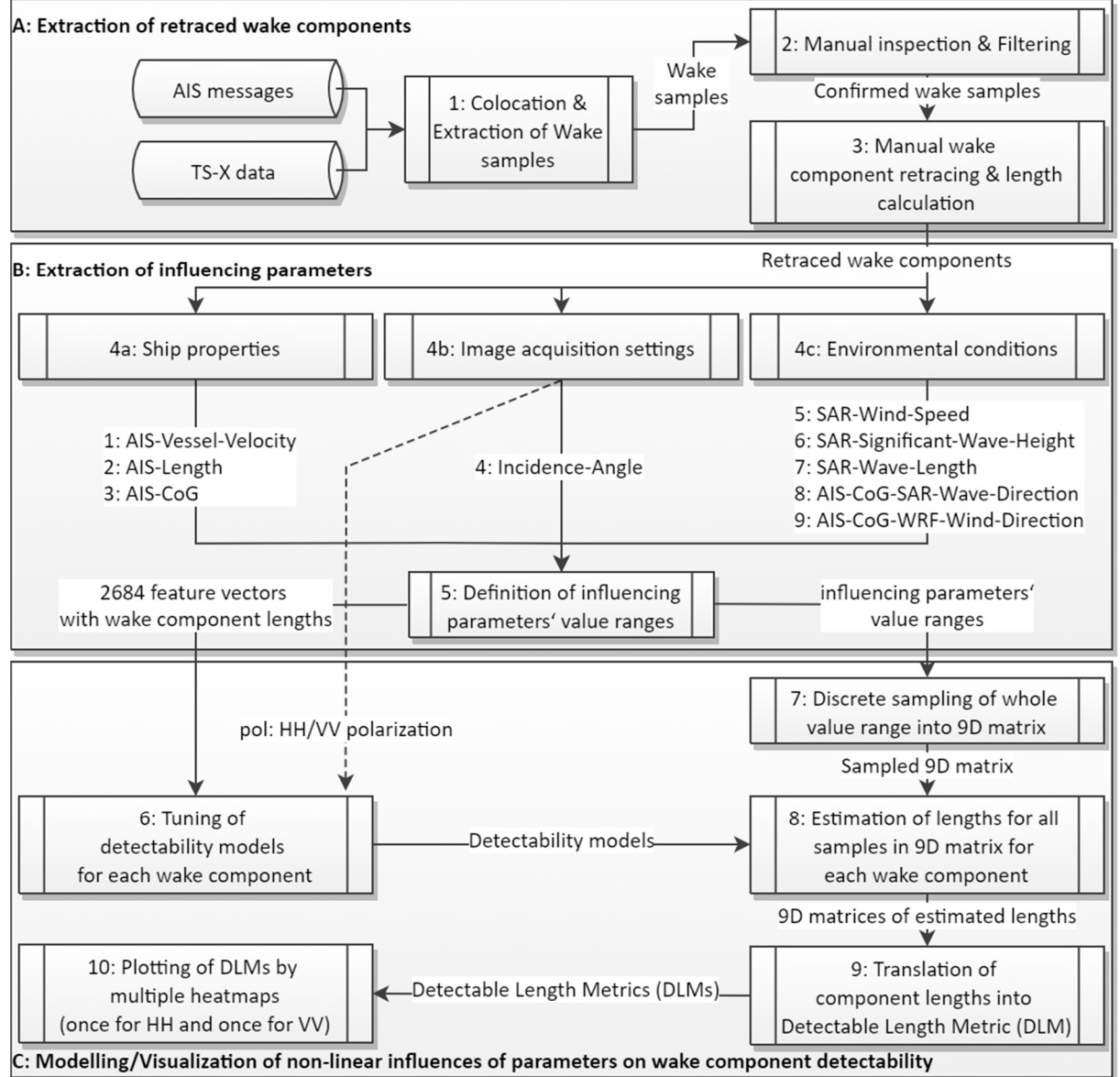

**Figure 9.** Flow-chart describing the overall process from data preparation (**A**) over retrieval of relevant parameter influencing the detectability (**B**) up to building of the detectability model and its visualization (**C**).

### 2.1. Extraction of Retraced Wake Components

In this study, a dataset consisting of around 800 high resolution TerraSAR-X images is used. The images were acquired within the years 2013 to 2017 over the North, Baltic and Mediterranean Seas. Most of the acquisitions were taken in Stripmap mode with HH-

polarization, but also Spotlight mode as well as VV-polarization or dual-co-polarization images are included.

Data from the Automatic Identification System (AIS) create the ground truth for this study. Therefore, the first step (Figure 9A1) executed for this study was the colocation of AIS and SAR data in space and time to extract a sample of candidate ship wakes. The colocation includes a correction of the ship displacement in the azimuth direction due to the ship motion in range direction [34,35] as well as an interpolation of the ship track between positions of transmitted AIS messages. After extraction, a manual inspection of all candidates was conducted (Figure 9A2). In this second step all examples were discarded, which show anomalies on the ocean surface with similar appearance as wakes, whose nature could not uniquely be explained by the motion of a ship on the ocean surface.

Then, on the remaining wake candidates a manual retracing of the wake components with curved appearance (i.e., near-hull turbulence, turbulent wake, Kelvin wake arms, V-Narrow wake arms and ship-generated internal waves) was executed (Figure 9A3). Wake components consisting of two curves propagating on the port and starboard side of the vessel were retraced separately. The presence of divergent or transverse waves was only flagged by "detected" or "not detected", because a retracing is not possible due to their oscillating nature. For the retraced wake components, the length was then calculated. In the whole training dataset, not a single example exists where all components are detectable. Around 21% of cases do not show any of the wake components. For wake components not present in the image, a component length of zero is set.

The final collection for detectability modeling contains 2684 wake samples. The proportions of wake components in the dataset differ strongly, as the overall detectability of the wake components also differs strongly. The respective proportions are listed in Table 1.

**Table 1.** Proportions of wake components in the wake component collection with 2684 wake samples.

| Wake Component | Proportion (Rounded to Integer) |
| --- | --- |
| Near-hull Turbulence | 60% |
| Turbulent wake | 62% |
| Port Kelvin wake arm | 20% |
| Starboard Kelvin wake arm | 20% |
| Port V-narrow wake arm | 28% |
| Starboard V-narrow wake arm | 25% |
| Port ship-generated internal wave | <1% |
| Starboard ship-generated internal wave | <1% |
| Transverse waves | 4% |
| Divergent waves | 6% |

*2.2. Extraction of Influencing Parameters*

Three categories of influencing parameters considered in this study are extracted in the fourth step (Figure 9B4a–4c). Most influencing parameters are based on environmental conditions (Figure 9B4c). It is preferable to retrieve the environmental conditions directly from the SAR images to account for local spatial and temporal variations. Therefore, the XWAVE_C [36–38] empirical algorithm is applied to derive sea state conditions from the ocean backscatter around the wakes. For estimation of the local wind speed in the vicinity of the wakes the XMOD-2 function [39,40] is applied. As XMOD-2 only provides the wind speed using a pre-calculated wind direction, the Weather Research and Forecasting Model (WRF) [41] is used for providing wind direction. SAR-Wind-Speed and SAR-Significant-Wave-Height are correlated [28], so as a combined measure of both influencing parameters, the Beaufort scale [42] is used in Section 3. All influencing parameters describing the corresponding ship's properties are derived from AIS (Figure 9B4a). From the image acquisition settings, only the incidence angle is considered here (Figure 9B4b).

In [12] it is described how the Pearson product–moment correlation coefficient and the parameter's redundancy are compared in order to select a set of nine influencing

parameters for further analysis. The parameter selection process was not repeated for this study. Instead the same set of nine influencing parameters is adopted. The parameters are listed in Table 2 together with brief descriptions. Table 2 also lists value ranges and default settings for plotting of results (Figure 9B5).

**Table 2.** List of the nine influencing parameters considered in the detectability model along with a description, the value range and a default parameter setting used for the plots in Section 3.

| Nr $i$ | Influencing Parameter Name ($x_i$) | Description | Value Range (*Default Setting*) |
|---|---|---|---|
| 1 | AIS-Vessel-Velocity ($x_1$) | Velocity of the vessel derived from AIS messages interpolated to the image acquisition time | 1 m/s to 10 m/s (6 m/s) |
| 2 | AIS-Length ($x_2$) | Length of the corresponding vessel based on AIS information | 5 m to 350 m (100 m) |
| 3 | AIS-CoG ($x_3$) | The CoG based on AIS information relative to the radar looking direction (0° means parallel to range and 90° mean parallel to Azimuth). [1] | 0° to 90° (45°) |
| 4 | Incidence-Angle ($x_4$) | Incidence angle of the radar cropped to TS-X's full performance value range | 20° to 45° (30°) |
| 5 | SAR-Wind-Speed ($x_5$) | Wind speed estimated from the SAR background around the vessel using the XMOD-2 geophysical model function | 2 m/s to 9 m/s (6 m/s) |
| 6 | SAR-Significant-Wave-Height ($x_6$) | Significant wave height estimated from the SAR background around the vessel using the XWAVE_C empirical algorithm | 0 m to 2 m (0.5 m) |
| 7 | SAR-Wave-Length ($x_7$) | Wavelength estimated from the SAR background around the vessel using the XWAVE_C empirical algorithm | 75 m to 350 m (150 m) |
| 8 | AIS-CoG-SAR-Wave-Direction ($x_8$) | Absolute angular difference between AIS-CoG and wave direction estimated from the SAR background around the vessel using the XWAVE_C empirical algorithm. [1] | 0° to 90° (45°) |
| 9 | AIS-CoG-WRF-Wind-Direction ($x_9$) | Absolute angular difference between AIS-CoG and wind direction estimated by the Weather Research and Forecasting Model (WRF) nearby the vessel. [1] | 0° to 90° (45°) |

[1] The 0°–360° value range of this parameter has been projected to 0°–90° as displayed in Figure 10.

### 2.3. Modelling of Non-Linear Influences of Parameters on Wake Component Detectability

Input to the modelling procedure is the training dataset of wake samples, where each wake sample is characterized by a feature vector of nine influencing parameters. In addition, each wake sample stores the measured length of each wake component (in meter), except for transverse and divergent waves. For transverse and divergent waves only the binary flag is included, like it was done in [11,12]. The datasets have been uploaded as Supplementary Material to this publication for reproductivity of the results.

HH-polarized images are expected to be better suited than VV-polarization for detecting Kelvin wakes [9,31], but ocean surface films, as substitute for turbulent wakes, show no significant difference in detectability between both polarizations [11,20,26]. As for Kelvin wakes, only the general detectability is increased in VV-polarized images, but the influence of the parameters on the detectability is not affected, in this study HH- or VV-polarization is taken into account by adding an additional nominal attribute to the feature vector of influencing parameters, which defines the polarization under which the respective wake sample has been acquired.

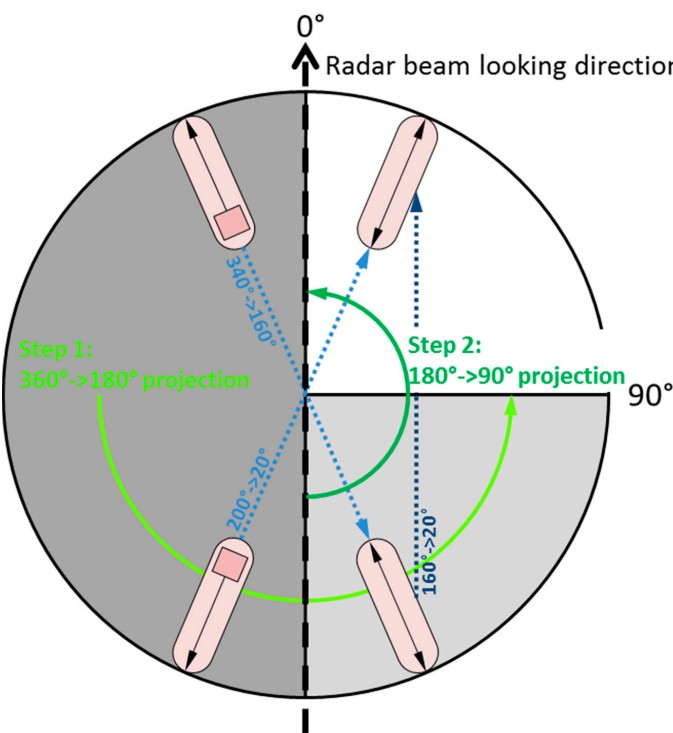

**Figure 10.** Example for a projection of a ship's heading from 0°–360° down to 0°–90°. The same projection has been applied to all influencing parameters measured in degrees.

Support Vector Machine (SVM) models have the advantage over other machine learning methods that they can be easily tuned in their complexity [43]. The model's complexity is of major importance in this study, as the relationship between the influencing parameters and the wake component's detectability is expected simple. Additionally, the datasets are noisy and overfitting must be prevented. SVMs have already proven beneficial in previous studies, so the model developed here is also based on SVMs [11,12,44]. With the prediction parameter, the wake component length, being now a continuous variable instead of a binary class (as in [11,12]), SVM regression is used instead of binary SVM classification [45]. This means one Support Vector Regression (SVR) model is trained for each wake component using the nine influencing parameters plus the nominal polarization attribute as features and the wake component length as prediction parameter (Figure 9C6). For both, SVM classification and SVM regression the LibSVM library was used [45].

SVMs can be parameterized by a variety of hyperparameters [45]. The most important hyperparameters are the settings of the kernel function. Here, the polynomial kernel with second degree polynomial is selected for two reasons:

1.  All statements about wake component detectability derived from simulations or theoretical studies, as recapped in Section 1.2, show either linear (first degree polynomial) or second-degree polynomial dependency to the influencing parameters. Note that this is valid, as long as the influencing parameters with units measured in degree are projected from the $[0°, 360°)$ to the $[0°, 90°)$ value range as described in Figure 10, otherwise two peaks (for $[0°, 180°]$ value range) or four peaks (for $[0°, 360°]$ value range) would be possible.
2.  In the previous study [12] it was discovered that a polynomial kernel higher than second degree and generally the radial-basis or sigmoid kernel lead to overfitting. As the model here has to solve a more complex problem, i.e., regression instead of binary classification, and the amount of data has not been increased, overfitting would be even more likely with increased model complexity [43].

The lowest possible Gamma-hyperparameter of zero was set to allow the SVM hyperplanes a strong curvature. As the model's complexity is already restricted by the low degree

of the polynomial kernel, it was assumed beneficial to increase the model's complexity by this means so that sharper peaks might be learned.

The other hyperparameters of the SVMs were tuned on the basis of 10-fold-cross-validation to quantify the model's performance [46]. The hyperparameters performing best on most of the wake component datasets are given in Table 3. More details about the tuning of hyperparameters and an explanation of the parameters can be found in [12,47].

**Table 3.** Settings of the Support Vector Machine (SVM)'s hyperparameters achieving highest 10-fold-cross validation accuracy on the training dataset.

| Hyperparameter Name | Value |
| --- | --- |
| SVM type | Epsilon-SVR |
| Kernel type | polynomial |
| Kernel degree | 2 |
| Epsilon loss | 0.1 |
| Cost | 1.0 |
| Gamma | 0.0 |
| Coef0 | 100 |

The detectability of the following wake components is modelled by the described SVM regression approach: near-hull turbulence ($ht$), turbulent wake ($tw$), port Kelvin wake arm ($pk$), starboard Kelvin wake arm ($sk$), accumulated port and starboard Kelvin wake arm ($psk$), port V-narrow wake ($pv$), startboard V-narrow wake ($sv$), accumulated port and starboard V-narrow wake ($psv$), ship-generated port internal wave wake ($pi$), ship-generated starboard internal wave wake ($si$) and accumulated ship-generated port and starboard internal wave wake ($psi$). Accumulation of port and starboard wake components means that an additional dataset is created by summarization of port and starboard wake component lengths. In the following formalization the set of wake components is denoted; $W = \{ht, tw, pk, sk, \ psk, pv, sv, \ psv, pi, si, \ psi\}$. In theory, each trained model $f_w$ is capable of estimating the detectable length $l_w$ of a certain wake component $w \in W$ in a SAR image given information about environmental conditions, ship properties and image acquisition settings:

$$l_w = f_w \left( x_1, x_2, x_3, x_4, x_5, x_6, x_7, x_8, x_9, x_{pol} \right) \tag{2}$$

where $x_i \ \forall i \in \{i \in \mathbb{N} \mid 1 \leq i \leq 9\}$ denotes one of the nine influencing parameters listed in Table 2 using the subscript $i$ as index and $x_{pol}$ is the nominal polarization attribute.

The generation of a nine-dimensional non-linear detectability model is only possible when the dataset contains enough wake samples with the respective wake component being detectable. For ship-generated internal waves, the trained models always predicted a component length close to zero. With over 99% of samples having an actual component length of zero in the dataset (see Table 1), the model already achieves an accuracy of 99% with this behavior. Thus, it was concluded that not enough information is available for modelling the detectability of ship-generated internal waves. For the sake of completeness, in the results chapter (Section 3) only a brief qualitative analysis of detectability of internal waves can be provided.

For transverse waves, divergent waves, Kelvins wake arms and V-narrow wake arms two datasets are available, respectively. One dataset contains information about the port side signatures and the other dataset information about the starboard side signatures. By comparison of the results from port and starboard side detectability models, respectively, it turned out that the models for Kelvin wake arms and V-narrow wake arms are similar, while the models for transverse waves and divergent waves showed discrepancies. The transverse waves and divergent waves mainly have also detectable port side signatures, when the starboard side signatures are detectable. In contrast, Kelvin wake arms and

V-narrow wake arms are often only detectable on one side. Therefore, it was concluded that enough data is available for modelling Kelvin wake arms and V-narrow wake arms and in turn also for modelling near-hull turbulences and turbulent wakes (as even more data is available for the latter two, see Table 1). The model complexity was reduced to five dimensions for modelling the detectability of divergent waves and transverse waves. Subsequently, a comparison of port and starboard side detectability models also showed similar results for these two wake components.

In the five-dimensional models, all sea state related parameters, i.e., SAR-Significant-Wave-Height, SAR-Wave-Length and AIS-CoG-SAR-Wave-Direction, and the WRF-based wind direction parameter, i.e., AIS-CoG-WRF-Wind-Direction, were excluded. SAR-Wave-Length and AIS-CoG-SAR-Wave-Direction are only byproducts and not the objective of the XWAVE_C algorithm. Further, in [12] it was already stated that SAR-Wave-Length, AIS-CoG-SAR-Wave-Direction and AIS-CoG-WRF-Wind-Direction are affected by high noise. SAR-Significant-Wave-Height is known for its high correlation with SAR-Wind-Speed and its exclusion does not considerably reduce the expressiveness of the resulting detectability model.

This means that the results presented in the Section 3 for transverse and divergent wave's detectability are only based on the five influencing parameters: AIS-Vessel-Velocity, AIS-Length, SAR-Wind-Speed, Incidence-Angle and AIS-CoG. As only a binary flag is available for both wake components, another reduction in the detectability model's complexity is the application of SVM classification instead of SVM regression. This implies that the detectability models are based on binary classification and the probability of class affiliation to the class "detected" is taken as a figure of merit for probability of detection. This approach is similar to the detectability models created for whole wake signatures in [11] and [12], but with the set of five influencing parameters plus the nominal polarization attribute used as input and the setting of hyperparameters as specified in Table 3.

By comparison of results based on datasets with port, starboard and accumulated port and starboard wake components the robustness of models is checked. Robustness is crucial in this study, as the nature of models provides the actual information here and not the model's accuracy itself. Therefore, the robustness was double checked by changing the model's hyperparameters to settings with less accuracy performance estimated by 10-fold cross validation. Additionally, from each wake component dataset two reduced datasets were derived by filtering certain samples:

- A light filtering of all samples with values of wake component length below the 10th percentile and above the 90th percentile.
- A strong filtering of all samples with values of wake component length, AIS-Vessel-Velocity, AIS-Length, Incidence-Angle, SAR-Wind-Speed, SAR-Significant-Wave-Height and SAR-Wave-Length below the 10th percentile and above the 90th percentile.

While the magnitude of influences differs for the varying settings and datasets, the characteristics of influences are similar. However, non-robust results exist and are marked accordingly in the Section 3.2–3.8. Robustness is further discussed in Section 4.

### 2.4. Visualization of Non-Linear Influences of Parameters on Wake Component Detectability

The detectability can be visualized by so called heatmaps. In one heatmap only two influencing parameters can be plotted at a time. By plotting and arranging multiple heatmaps with varying settings of the respective other influencing parameters, the direct dependencies and interdependencies of parameters with respect to detectability, which are reproduced by the model, can be analyzed. The process of plotting and arranging the heatmaps has to be repeated multiple times to obtain a more complete overview of the model.

By discrete sampling of the whole value range of all nine influencing parameters and then defining a feature vector for each combination of those discrete values, a nine-dimensional matrix is obtained, which includes each possible discrete input to the detectability model (Figure 9C7). In theory, the input to the detectability model can be a feature vector with any non-discrete values. In order to restrict the number of possible

inputs and to define a nine-dimensional matrix with finite size, the value range of each influencing parameter listed in Table 2 is discretized in a way so that a maximum number of discrete samples per influencing parameter exists, but still the step size is small enough to represent the variations in detectability. As an example, for Incidence-Angle 26 discrete integer steps are used (i.e., 26 steps from 20° to 45°) and 15° is the step distance used for parameters based on course over ground (i.e., 7 steps: 0°, 15°, 30°, 45°, 60°, 75°, 90°). In total, 87 bins are used.

Each feature vector is then used as input to the detectability model of each wake component. Each model then estimates the respective wake component lengths, which should theoretically be detectable given the varying conditions defined by the input feature vectors (Figure 9C8). The result is a collection of multiple 9D matrices, where each matrix contains the estimated wake component lengths for the respective wake component based on all possible discrete inputs. The process is executed twice, once for HH polarization and once for VV polarization.

The wake component length is used here as an indicator for the wake component detectability. In the following a figure of merit is introduced in order to normalize this indicator for wake component detectability for the various models and wake components, respectively. A normalized figure of merit also facilitates the visualization of wake component detectability. This figure of merit is denoted: Detectable Length Metric $DLM_w$. The $DLM_w$ represents an expected detectable wake component length under certain conditions and should not be confused with probability of detection. In contrast, the binary classification model used for detectability analysis of transverse and divergent waves really provides a probability, but this probability is also just an indication for probability of detection.

In the style of a probability of detection for the detectable length metric a value range of $DLM_w \in [0,1]$ is defined, where $DLM_w = 0$ implies that the wake component $w$ is not detectable and $DLM_w = 1$ implies maximum detectability of the wake component $w$. The normalization is achieved by scaling the estimated wake component lengths within a minimum and maximum length boundary (Figure 9C9). In theory, the minimum boundary should be set to zero meters for any wake component, as only wake components with zero length would be always undetectable. However, there are other factors, which need to be taken into account for each wake component specifically, so that also wake component lengths above zero meters can mean that the wake component is not detectable. For example, most wake components originate from the ship's bow and therefore their length is also measured beginning from the bow. The ship itself also has a size and moreover the ship's SAR signature often is imaged to an even larger extent in the SAR images than the ship's extent is in reality [1]. Thus, the region near the wake vortex cannot contain any usable information about the ocean surface, because the radar signal is already reflected from the ship hull without any proportion reaching the ocean surface. Moreover, the oversized imaging of the ship signature often interferes with the much weaker information reflected from the ocean surface, making a detection of wake components in the area impossible. Additionally, some wake components are dominant over others when they are present. The maximum length boundary defines that all length values equal or above the boundary imply a definite detection of the respective wake component. Therefore, to obtain $DLM_w$ each estimated wake component length is linearly normalized between the respective wake components minimum length boundary $l_w^{min}$ and maximum length boundary $l_w^{max}$ by:

$$DLM_w = \frac{l_w - l_w^{min}}{\left| l_w^{max} - l_w^{min} \right|} \tag{3}$$

Finally, to generate the detectability heatmaps for a certain wake component, the respective 9D matrix of component length estimates needs to be read out by accessing the estimated length value according to the desired settings of influencing parameters for which the heatmap is being plotted. As the heatmaps can only visualize two influencing parameters on the two axes, all other parameters are set to fixed values for the respective plot. In Table 2 default parameter settings are defined (in italic letters), which are used

during the plotting, when no other fixed values are specified. After normalizing the length based on the wake component's specific minimum and maximum length boundary, a color code is assigned to the obtained $DLM_w$. Then the color is displayed in the form of a rectangle at its position corresponding to the X- and Y-axis labels of the heatmap (Figure 9C10). A full presentation of all generated heatmaps, which were analyzed during the evaluation process, in one paper is impossible. Therefore, only a selection of heatmaps is presented in Section 3 to support the derived statements.

## 3. Results

The goal of this study is gaining knowledge about the impact of the influencing parameters on the detectability of certain wake components. For whole wakes the characteristics of these influences have been defined in [12] and categorized into four types. In Section 3.1, the characteristics are briefly described with respect to single wake components instead of whole wakes. Then, the influencing parameters are categorized according to these characteristics in the Section 3.2–3.8. A selection of heatmaps is added so that the categorizations can be understood.

For easier understanding of how to interpret the heatmaps, Figure 11 is now partially explained as an example: The influencing parameters AIS-Vessel-Velocity, AIS-Length and AIS-CoG are variable in Figure 11, while all other parameters are set to their default values as specified in Table 2. Each combination of the variable parameters AIS-Vessel-Velocity, AIS-Length and AIS-CoG produce different model responses, when given as input to wake detectability model for near-hull turbulences. For AIS-Vessel-Velocity $x_1 = 7$ m/s, AIS-Length $x_2 = 20$ m and AIS-CoG $x_3 = 45°$ the respective seventh column and fourth row in the left plot shows an orange color. From the colorscale on the right a $DLM_w \approx 30$ can be read out. As the other colors in the same column are all redder than for this fourth row, the $DLM_w \approx 30$ is the maximum. This implies a one-peaked maximum influence of AIS-CoG for $x_1 = 7$ m/s and $x_2 = 20$ m. As already described in [11], the characteristics do rarely differ between HH- and VV- polarization. Therefore, the selected heatmaps are by default visualizing the detectability based on HH-polarization. However, for some influencing parameters differences were encountered between characteristics due to different polarization. In these cases, the characteristics are described for HH- and VV-polarization and both heatmap versions are presented in the corresponding figures.

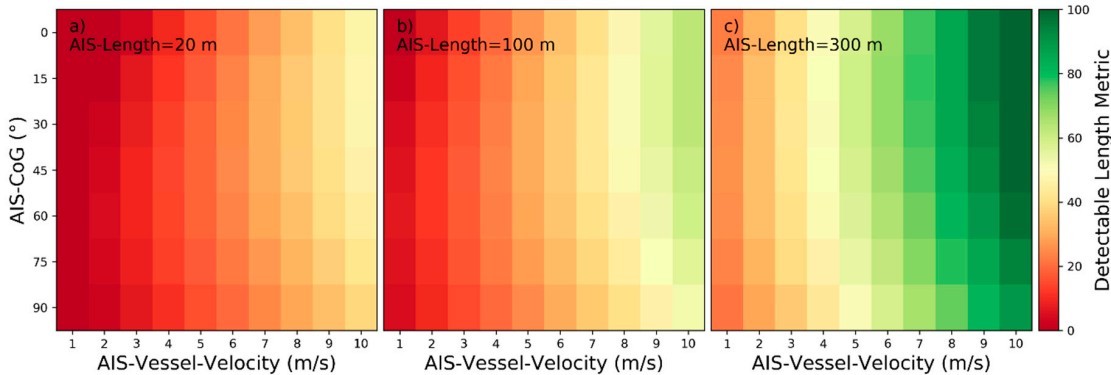

**Figure 11.** Detectability heatmaps for near-hull turbulences based on AIS-Vessel-Velocity, AIS-CoG and from left to right AIS-Length with (**a**) 20 m, (**b**) 100 m, and (**c**) 300 m.

### 3.1. Characteristics of Influences on Wake Component Detectability

The following characteristics are valid only for the defined value ranges as defined in Table 2. The influencing parameter with index $c$, for which the characteristics are described, is denoted here as $x_c$ and its value range is denoted as $I_c$. The respective other parameters $x_o$ with indices $o \in \{i \in \mathbb{N} \,|\, 1 \leq i \leq 9 \wedge i \neq c\}$ are in the set $X_{o \neq c}$ and their corresponding value ranges are denoted $I_o$.

1. Influencing parameters with no influence on wake component detectability:

$$f_w{}'\left(x_c, X_{o \neq c}\right) = \frac{\partial f_w}{\partial x_c} = 0, \ \forall x_c \in I_c, \ \forall x_o \in I_o \tag{4}$$

and

$$f_w{}''\left(x_c, X_{o \neq c}\right) = \frac{\partial f_w{}'}{\partial x_c} \neq 0, \ \forall x_c \in I_c, \ \forall x_o \in I_o \tag{5}$$

2. Influencing parameters with independent monotonic influence on wake component detectability:

$$f_w{}'\left(x_c, X_{o \neq c}\right) = \frac{\partial f_w}{\partial x_c} \lessgtr 0, \ \forall x_c \in I_c, \ \forall x_o \in I_o \tag{6}$$

and

$$f_w{}''\left(x_c, X_{o \neq c}\right) = \frac{\partial f_w{}'}{\partial x_c} \neq 0, \ \forall x_c \in I_c, \ \forall x_o \in I_o \tag{7}$$

3. Influencing parameters with a one-peaked maximum or minimum influence on wake component detectability at $x_{c,peak}$:

$$f_w{}'\left(x_{c,peak}, X_{o \neq c}\right) = \frac{\partial f_w}{\partial x_c} = 0, \ \exists x_{c,peak} \in I_c, \ \forall x_o \in I_o \tag{8}$$

and

$$f_w{}''\left(x_{c,peak}, X_{o \neq c}\right) = \frac{\partial f_w{}'}{\partial x_c} \neq 0, \ \exists x_{c,peak} \in I_c, \ \forall x_o \in I_o \tag{9}$$

4. Influencing parameters with interdependent influence on wake component detectability:

When the magnitude combination of other influencing parameters reaches a certain range of settings $I_{o,turn}$, the influence of the characterized influencing parameter is not monotonic anymore, but shows a one-peaked maximum or minimum at $x_{c,peak}$. (see Equations (11) and (12)). The monotonic influence may even be reversed within the defined value range for further monotonically changing magnitudes of the other influencing parameters.

$$f_w{}'\left(x_c, X_{o \neq c}\right) = \frac{\partial f_w}{\partial x_c} \lessgtr 0, \ \forall x_c \in I_c, \ \forall x_o \in I_o \smallsetminus I_{o,turn} \tag{10}$$

with

$$f_w{}'\left(x_{c,peak}, X_{o \neq c}\right) = \frac{\partial f_w}{\partial x_c} = 0, \ \exists x_{c,peak} \in I_c, \ \exists x_o \in I_{o,turn} \tag{11}$$

and

$$f_w{}''\left(x_{c,peak}, X_{o \neq c}\right) = \frac{\partial f_w{}'}{\partial x_c} \neq 0, \ \exists x_{c,peak} \in I_c, \ \exists x_o \in I_{o,turn} \tag{12}$$

### 3.2. Categorization of Influencing Parameters by Characteristics of Influences for Near-Hull Turbulences

For near-hull turbulences the minimum length boundary is $l_t^{min} = 0$ m and the maximum length boundary is $l_t^{max} = 200$ m. Near-hull turbulences are detectable only in the close vicinity of the corresponding vessels and their lengths aft of the vessels are rarely exceeding the vessel's length. The maximum length boundary for the heatmap plots in Figures 11–17 corresponds to the near-hull turbulence length at the 80th percentile over all wake samples with measured near-hull turbulences. According to the plots, the nine influencing parameters can be categorized as described in Table 4.

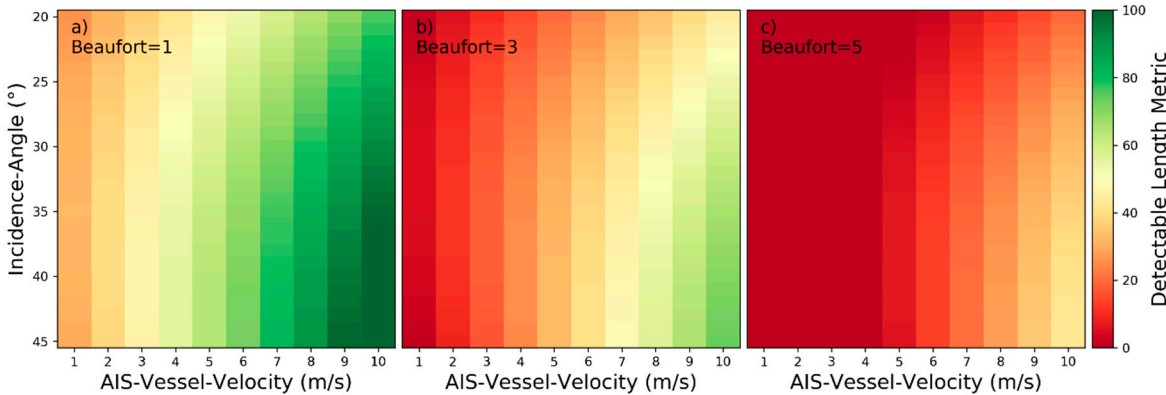

**Figure 12.** Detectability heatmaps for near-hull turbulences based on AIS-Vessel-Velocity, Incidence-Angle and from left to right Beaufort numbers with (**a**) 1 bft, (**b**) 3 bft, and (**c**) 5 bft.

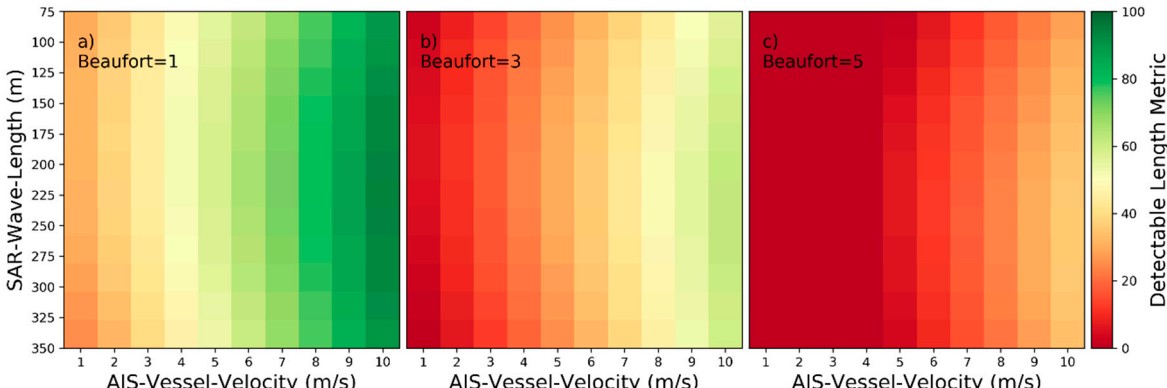

**Figure 13.** Detectability heatmaps for near-hull turbulences based on AIS-Vessel-Velocity, SAR-Wave-Length and from left to right Beaufort numbers with (**a**) 1 bft, (**b**) 3 bft, and (**c**) 5 bft.

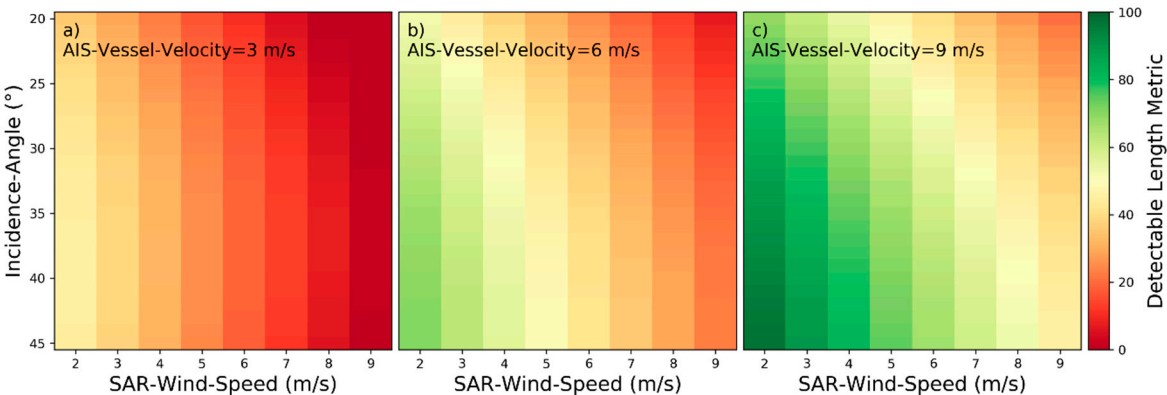

**Figure 14.** Detectability heatmaps for near-hull turbulences based on SAR-Wind-Speed, Incidence-Angle and from left to right AIS-Vessel-Velocity with (**a**) 3 m/s, (**b**) 6 m/s, and (**c**) 9 m/s.

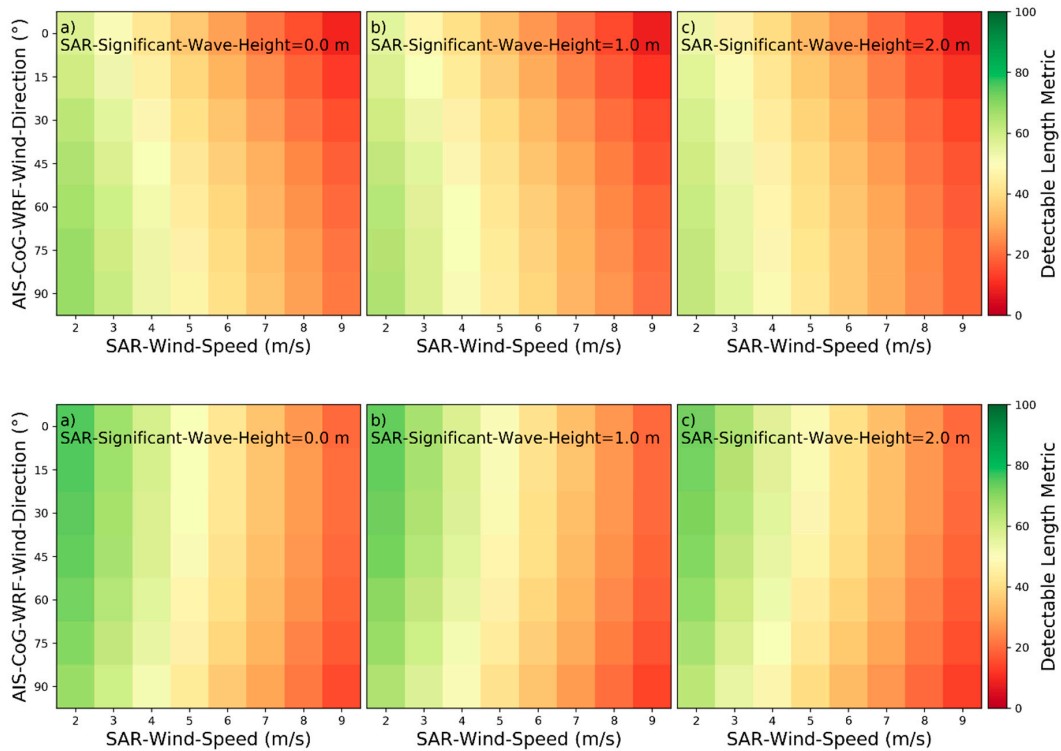

**Figure 15.** Detectability heatmaps for near-hull turbulences based on SAR-Wind-Speed, AIS-CoG-WRF-Wind-Direction and from left to right SAR-Significant-Wave-Height with (**a**) 0.0 m, (**b**) 1.0 m, and (**c**) 2.0 m (**top plots**: HH-polarization, **bottom plots**: VV-polarization).

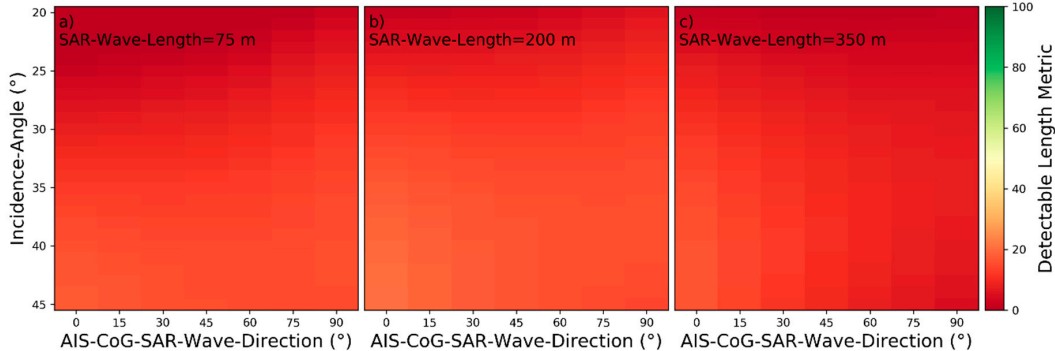

**Figure 16.** Detectability heatmaps for near-hull turbulences based on AIS-CoG-SAR-Wave-Direction, Incidence-Angle and from left to right SAR-Wave-Length with (**a**) 75 m, (**b**) 200 m, and (**c**) 350 m.

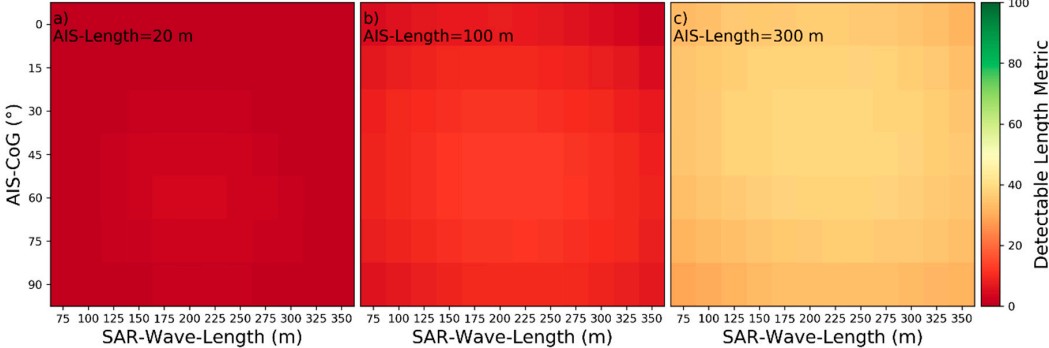

**Figure 17.** Detectability heatmaps for near-hull turbulences based on SAR-Wave-Length, AIS-CoG and from left to right AIS-Length with (**a**) 20 m, (**b**) 100 m, and (**c**) 300 m.

**Table 4.** Categorization of influencing parameters by characteristics of influences for near-hull turbulences.

| Characterized Influencing Parameter $x_c$ | Characteristic of Influence | Relevant Figure |
|---|---|---|
| AIS-Vessel-Velocity ($x_1$) | **Positive independent monotonic influence** $f_w'(x_c, X_{o \neq c}) > 0$ | Figure 11, Figure 12, Figure 13 and Figure 14 |
| AIS-Length ($x_2$) | **Positive independent monotonic influence** $f_w'(x_c, X_{o \neq c}) > 0$ | Figure 11 and Figure 17 |
| AIS-CoG ($x_3$) | Interdependent influence with AIS-Vessel-Velocity One-peaked maximum influence for and $f_w''(x_{c,peak}, X_{o \neq c}) < 0$ with $x_{c,peak} \approx 45°$, but neither pronounced nor robust, thus no influence $f_w'(x_c, X_{o \neq c}) \approx 0$ negative monotonic influence for $x_1 \geq 6 \text{ m/s}$, $f_w'(x_c, X_{o \neq c}) < 0$ (not robust) | Figure 11, (Figure 17) |
| Incidence-Angle ($x_4$) | **Interdependent influence** with AIS-Vessel-Velocity no influence for $x_1 \leq 3 \text{ m/s} f_w'(x_{c,peak}, X_{o \neq c}) = 0$ Positive monotonic influence for $x_1 > 3 \text{ m/s} f_w'(x_c, X_{o \neq c}) > 0$ | Figure 12, Figure 14 and Figure 16 |
| SAR-Wind-Speed ($x_5$) | **Negative independent monotonic influence** $f_w'(x_c, X_{o \neq c}) < 0$ | Figure 12, Figure 13, Figure 14 and Figure 15 |
| SAR-Significant-Wave-Height ($x_6$) | Negative independent monotonic influence $f_w'(x_c, X_{o \neq c}) < 0,$ but not pronounced, thus **no influence** $f_w'(x_c, X_{o \neq c}) \approx 0$ | Figure 15 |
| SAR-Wave-Length ($x_7$) | One-peaked maximum influence $f_w'(x_{c,peak}, X_{o \neq c}) = 0$ and $f_w''(x_{c,peak}, X_{o \neq c}) < 0$ with $x_{c,peak} \approx 200 \text{ m}$, but neither pronounced nor robust, thus **no influence** $f_w'(x_c, X_{o \neq c}) \approx 0$ | Figure 13, Figure 16 and Figure 17 |
| AIS-CoG-SAR-Wave-Direction ($x_8$) | Interdependent influence with SAR-Wave-Length and Incidence-Angle positive monotonic influence for $x_7 \leq 200 \text{ m}$ and $x_4 \leq 30° f_w'(x_c, X_{o \neq c}) < 0$ no influence for $x_7 > 200 \text{ m}$ and $x_4 \leq 30° f_w'(x_c, X_{o \neq c}) = 0$ negative monotonic influence for $x_7 > 200 \text{ m}$ and $x_4 > 30° f_w'(x_c, X_{o \neq c}) > 0$ Neither pronounced nor robust, thus **no influence** $f_w'(x_c, X_{o \neq c}) \approx 0$ | Figure 16 |
| AIS-CoG-WRF-Wind-Direction ($x_9$) | **Positive independent monotonic influence (HH)** $f_w'(x_c, X_{o \neq c}) > 0$ **Negative independent monotonic influence (VV)** $f_w'(x_c, X_{o \neq c}) < 0$ | Figure 15 |

### 3.3. Categorization of Influencing Parameters by Characteristics of Influences for Turbulent Wakes

For turbulent wakes the minimum length boundary is $l_t^{min} = 200 \text{ m}$ and the maximum length boundary is $l_t^{max} = 1750 \text{ m}$. By the minimum length boundary, the near-hull turbulence around the ship is discarded. The maximum length boundary for the heatmap plots in the Figures 18–23 corresponds to the turbulent wake length at the 80th percentile over all wake samples with measured turbulent wakes. According to the plots, the nine influencing parameters can be categorized as described in the Table 5.

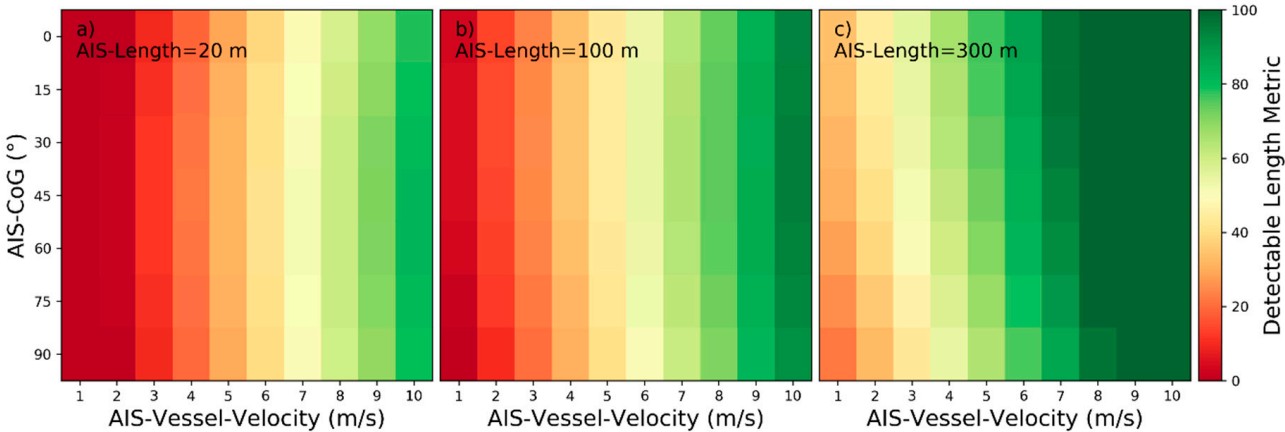

**Figure 18.** Detectability heatmaps for turbulent wakes based on AIS-Vessel-Velocity, AIS-CoG and from left to right AIS-Length with (**a**) 20 m, (**b**) 100 m, and (**c**) 300 m.

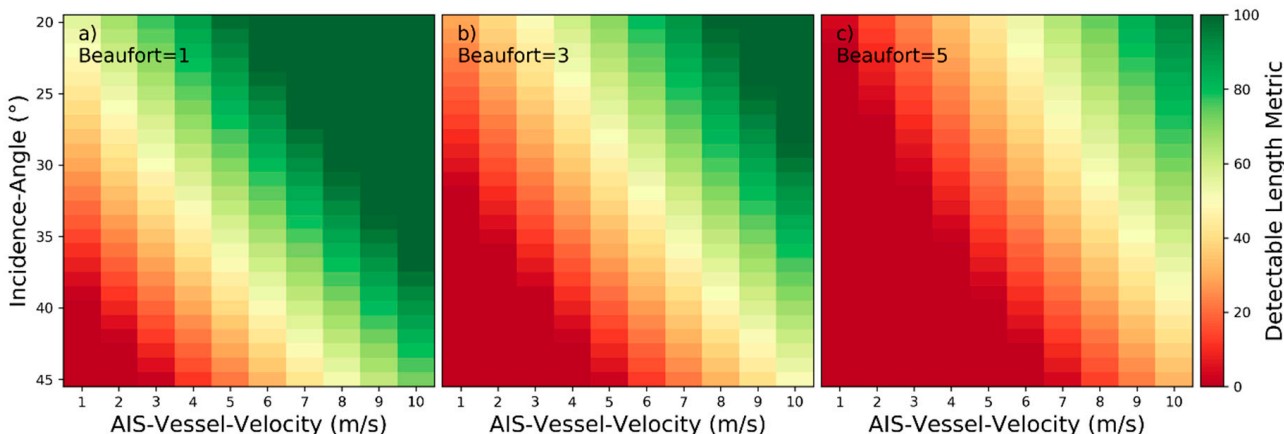

**Figure 19.** Detectability heatmaps for turbulent wakes based on AIS-Vessel-Velocity, Incidence-Angle and from left to right Beaufort numbers with (**a**) 1 bft, (**b**) 3 bft, and (**c**) 5 bft.

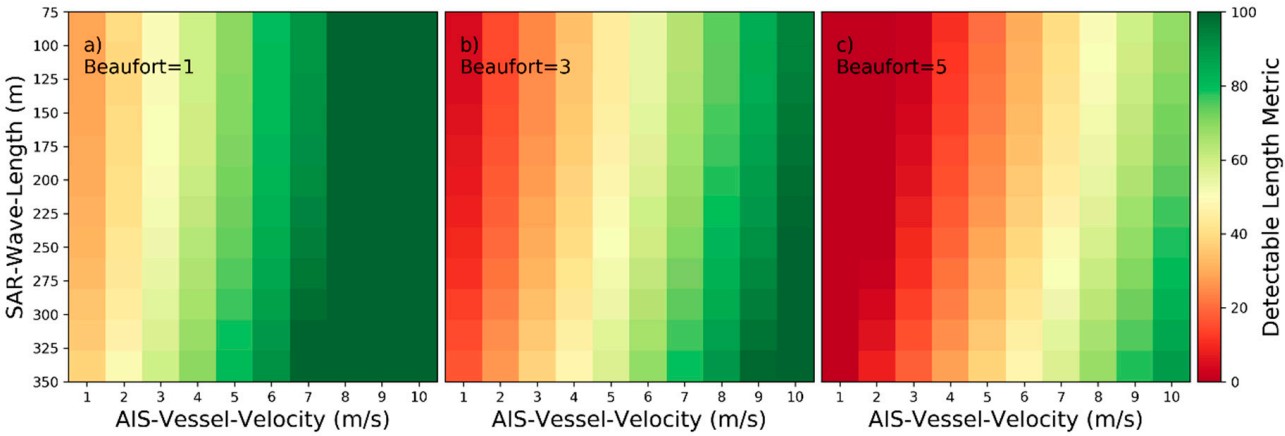

**Figure 20.** Detectability heatmaps for turbulent wakes based on AIS-Vessel-Velocity, SAR-Wave-Length and from left to right Beaufort numbers with (**a**) 1 bft, (**b**) 3 bft, and (**c**) 5 bft.

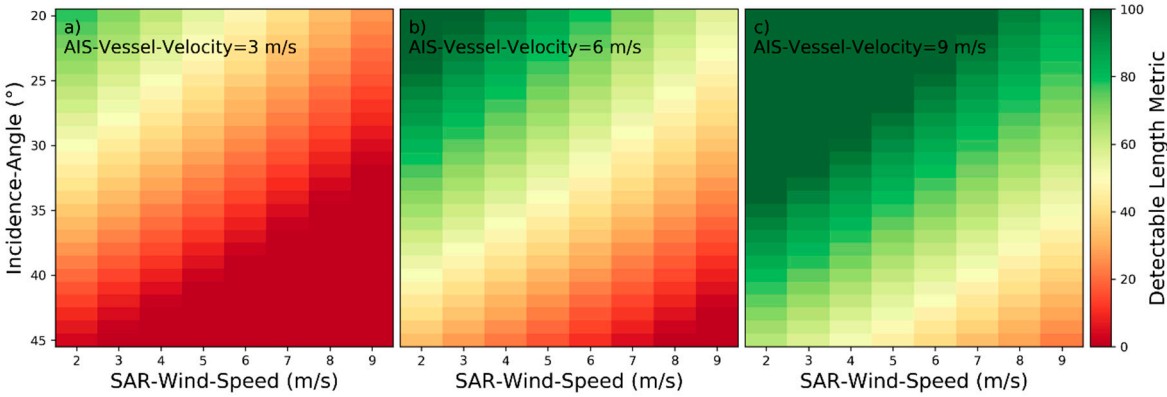

**Figure 21.** Detectability heatmaps for turbulent wakes based on SAR-Wind-Speed, Incidence-Angle and from left to right AIS-Vessel-Velocity with (**a**) 3 m/s, (**b**) 6 m/s, and (**c**) 9 m/s.

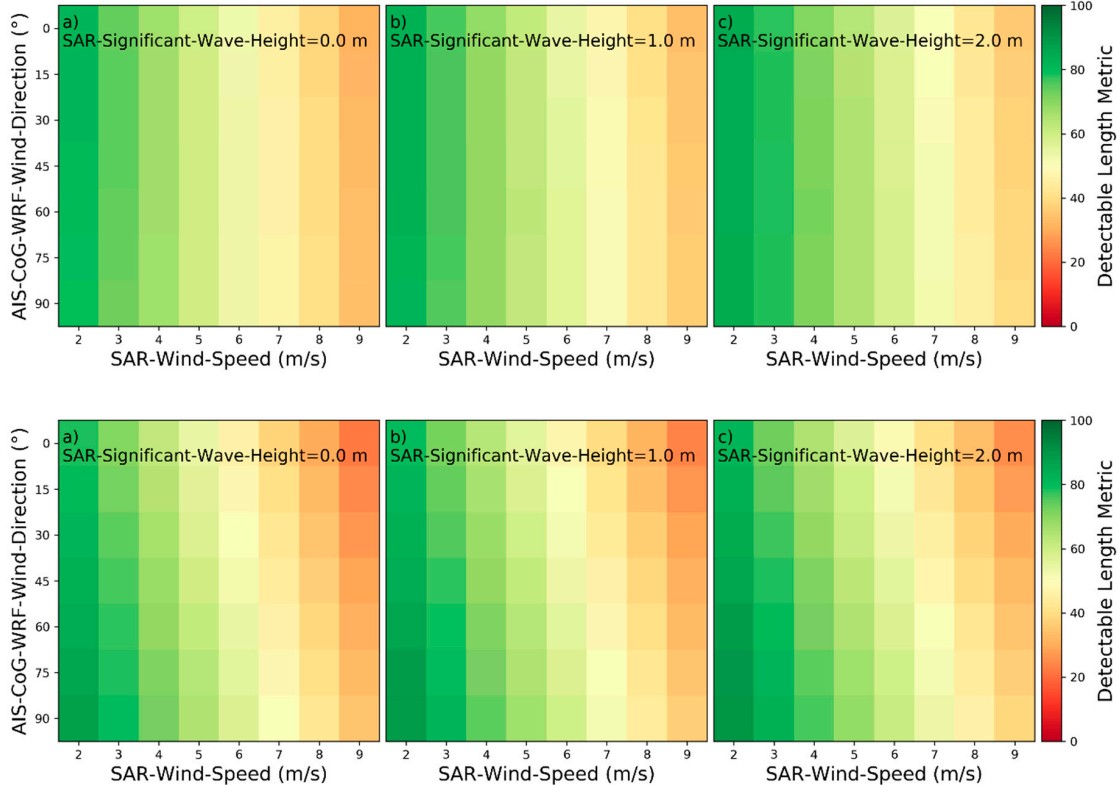

**Figure 22.** Detectability heatmaps for turbulent wakes based on SAR-Wind-Speed, AIS-CoG-WRF-Wind-Direction and from left to right SAR-Significant-Wave-Height with (**a**) 0.0 m, (**b**) 1.0 m, and (**c**) 2.0 m (**top plots**: HH-polarization, **bottom plots**: VV-polarization).

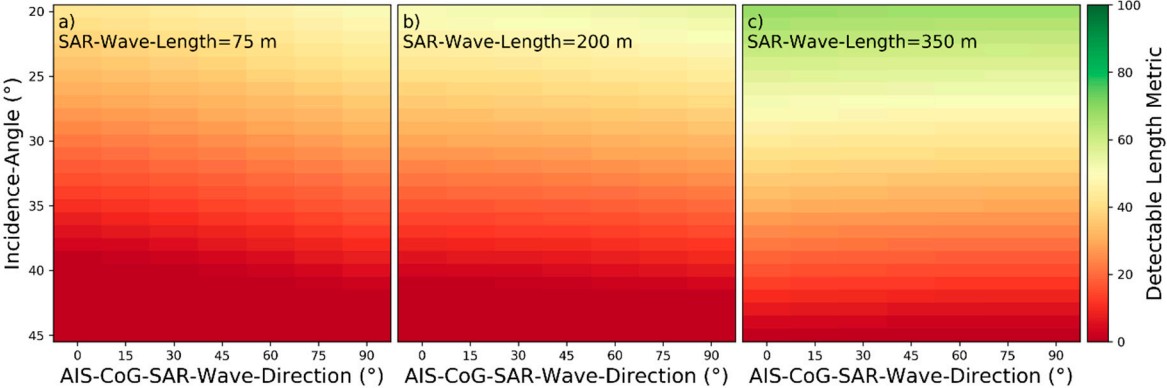

**Figure 23.** Detectability heatmaps for turbulent wakes based on AIS-CoG-SAR-Wave-Direction, Incidence-Angle and from left to right SAR-Wave-Length with (**a**) 75 m, (**b**) 200 m, and (**c**) 350 m.

**Table 5.** Categorization of Influencing Parameters by Characteristics of Influences for Turbulent Wakes.

| Characterized Influencing Parameter $x_c$ | Characteristic of Influence | Relevant Figure |
|---|---|---|
| AIS-Vessel-Velocity <br> $(x_1)$ | **Positive independent monotonic influence** <br> $f_w'(x_c, X_{o \neq c}) > 0$ | Figure 18, Figure 19, <br> Figure 20 and Figure 21 |
| AIS-Length <br> $(x_2)$ | **Positive independent monotonic influence** <br> $f_w'(x_c, X_{o \neq c}) > 0$ | Figure 18 and Figure 24 |
| AIS-CoG <br> $(x_3)$ | **Interdependent influence (HH)** <br> with SAR-Wave-Length <br> One-peaked maximum influence for <br> $x_7 \leq 200 \, \text{m} \, f_w'(x_{c,peak}, X_{o \neq c}) = 0$ and <br> $f_w''(x_{c,peak}, X_{o \neq c}) < 0$ with $x_{c,peak} \approx 45°$ <br> (neither pronounced nor robust) <br> Positive monotonic influence for <br> $x_7 > 200 \, \text{m} \, f_w'(x_c, X_{o \neq c}) > 0$ <br> **Negative independent monotonic influence (VV)** <br> $f_w'(x_c, X_{o \neq c}) > 0$ (not robust) | Figure 18 (only HH), Figure 24 |
| Incidence-Angle <br> $(x_4)$ | **Negative independent monotonic influence** <br> $f_w'(x_c, X_{o \neq c}) < 0$ | Figure 19, Figure 21 <br> and Figure 23 |
| SAR-Wind-Speed <br> $(x_5)$ | **Negative independent monotonic influence** <br> $f_w'(x_c, X_{o \neq c}) < 0$ | Figure 19, Figure 20, Figure 21 <br> and Figure 22 |
| SAR-Significant-Wave-Height <br> $(x_6)$ | Positive independent monotonic influence <br> $f_w'(x_c, X_{o \neq c}) > 0$ <br> but not pronounced, thus **no influence** <br> $f_w'(x_c, X_{o \neq c}) \approx 0$ | Figure 22 |
| SAR-Wave-Length <br> $(x_7)$ | **Positive independent monotonic influence** <br> $f_w'(x_c, X_{o \neq c}) > 0$ | Figure 20, Figure 23 <br> and Figure 24 |
| AIS-CoG-SAR-Wave-Direction <br> $(x_8)$ | **Interdependent influence** with SAR-Wave-Length <br> positive monotonic influence for $x_7 \leq 250$ m <br> $f_w'(x_c, X_{o \neq c}) < 0$ (not pronounced) <br> no influence for $x_7 > 250 \, \text{m} \, f_w'(x_c, X_{o \neq c}) = 0$ | Figure 23 |
| AIS-CoG-WRF-Wind-Direction <br> $(x_9)$ | **No influence (HH)** <br> $f_w'(x_c, X_{o \neq c}) = 0$ <br> **Positive independent monotonic influence (VV)** <br> $f_w'(x_c, X_{o \neq c}) < 0$ (not robust) | Figure 22 |

### 3.4. Categorization of Influencing Parameters by Characteristics of Influences for Kelvin Wake Arms

For Kelvin wake arms the minimum length boundary is $l_t^{min} = 200$ m and the maximum length boundary is $l_t^{max} = 1500$ m. By the minimum length boundary, the near-hull turbulence around the ship is discarded. The maximum length boundary for the heatmap plots in the Figures 25–31 roughly corresponds to the length of the port and starboard Kelvin wake arms at the 80th percentile over all wake samples with measured Kelvin wake arms. According to the plots, the nine influencing parameters can be categorized as described in the Table 6.

### 3.5. Categorization of Influencing Parameters by Characteristics of Influences for Divergent Waves

The plots in this subsection are based on the binary SVM classification model instead of the SVR. Therefore, no minimum and maximum length boundary needs to be defined, as the detectable length metrics are provided by the classifiers directly. For divergent waves, the detectability heatmaps are presented in the Figures 32–34 and based on those heatmaps, the five influencing parameters can be categorized as described in the Table 7.

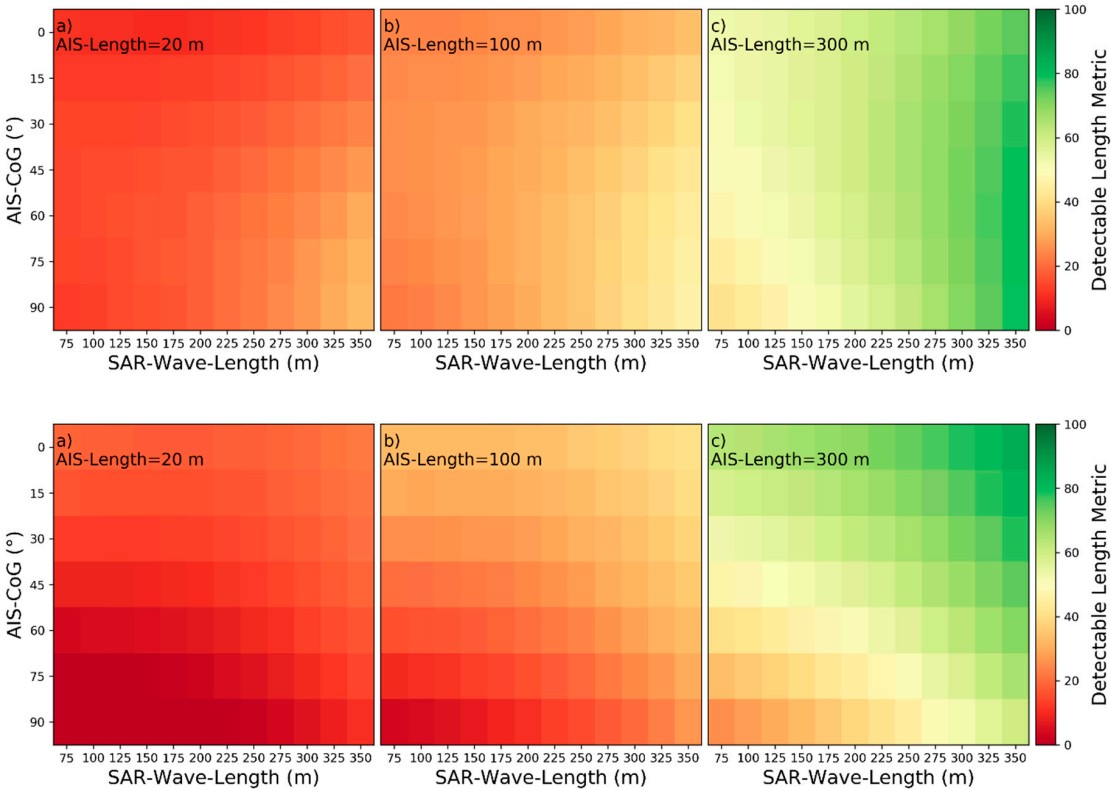

**Figure 24.** Detectability heatmaps for turbulent wakes based on SAR-Wave-Length, AIS-CoG and from left to right AIS-Length with (**a**) 20 m, (**b**) 100 m, and (**c**) 300 m (**top plots**: HH-polarization, **bottom plots**: VV-polarization).

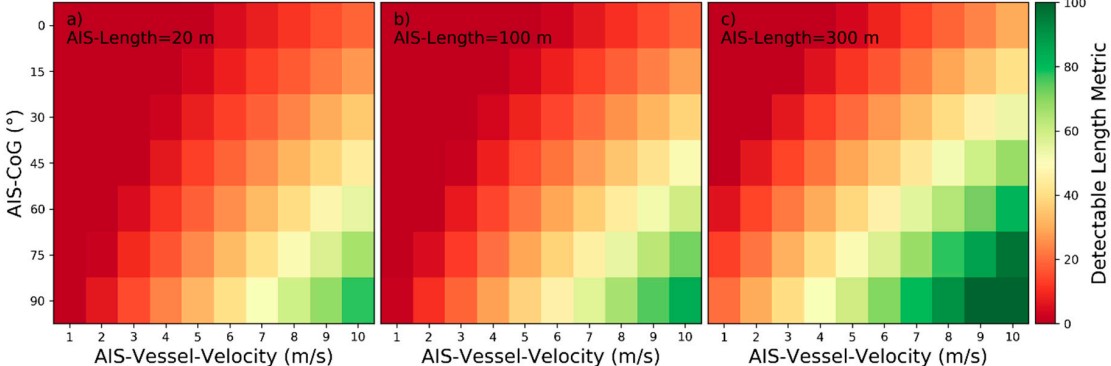

**Figure 25.** Detectability heatmaps for accumulated port and starboard Kelvin wake arms based on AIS-Vessel-Velocity, AIS-CoG and from left to right AIS-Length with (**a**) 20 m, (**b**) 100 m, and (**c**) 300 m.

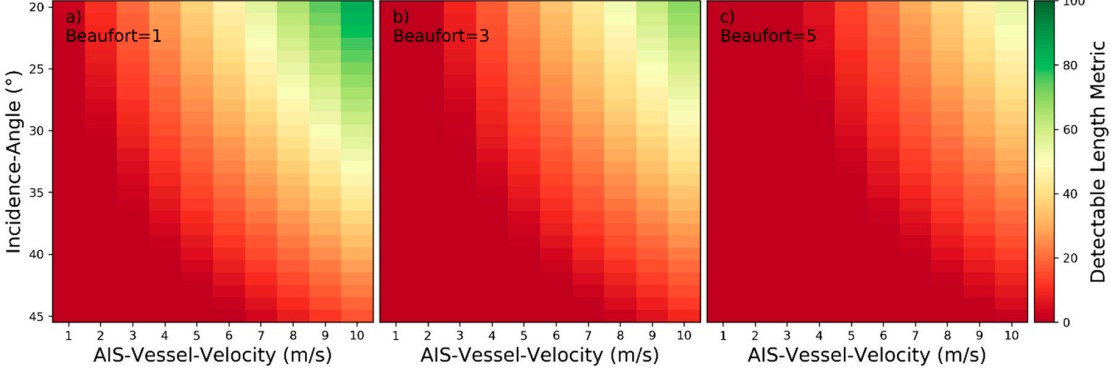

**Figure 26.** Detectability heatmaps for accumulated port and starboard Kelvin wake arms based on AIS-Vessel-Velocity, Incidence-Angle and from left to right Beaufort numbers with (**a**) 1 bft, (**b**) 3 bft, and (**c**) 5 bft.

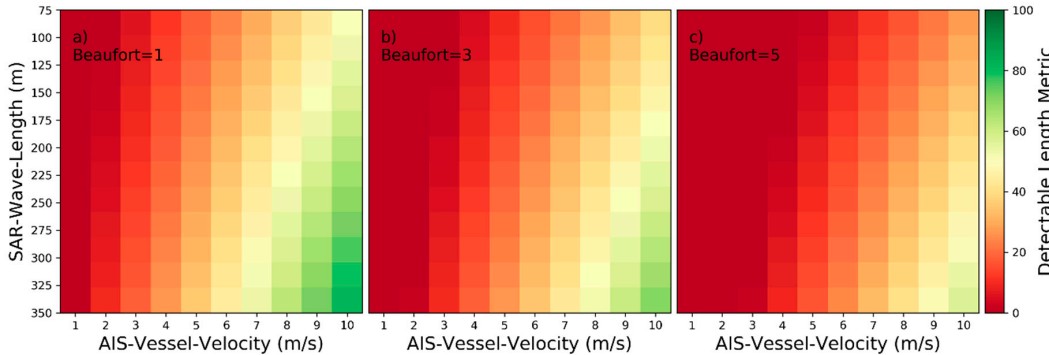

**Figure 27.** Detectability heatmaps for accumulated port and starboard Kelvin wake arms based on AIS-Vessel-Velocity, SAR-Wave-Length and from left to right Beaufort numbers with (**a**) 1 bft, (**b**) 3 bft, and (**c**) 5 bft.

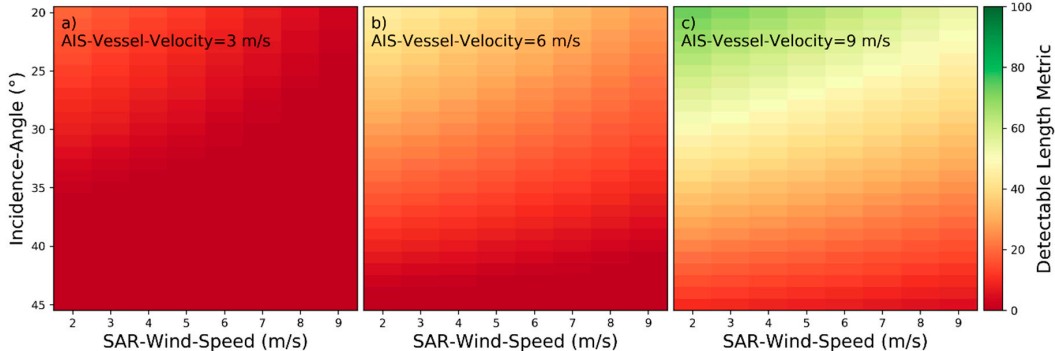

**Figure 28.** Detectability heatmaps for accumulated port and starboard Kelvin wake arms based on SAR-Wind-Speed, Incidence-Angle and from left to right AIS-Vessel-Velocity with (**a**) 3 m/s, (**b**) 6 m/s, and (**c**) 9 m/s.

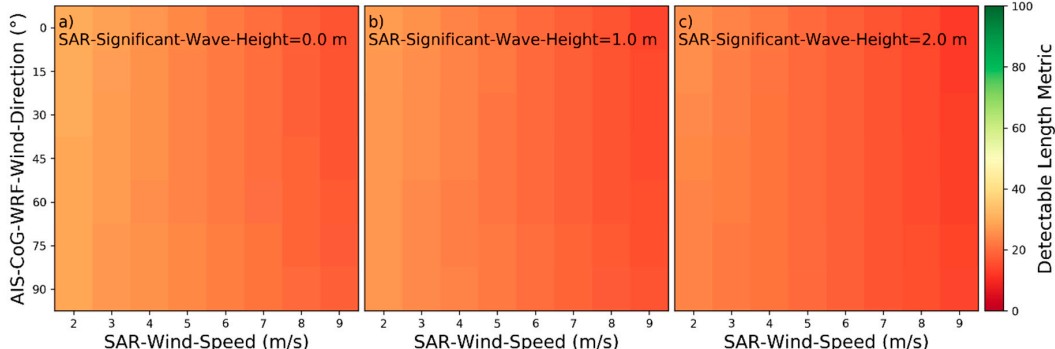

**Figure 29.** Detectability heatmaps for accumulated port and starboard Kelvin wake arms based on SAR-Wind-Speed, AIS-CoG-WRF-Wind-Direction and from left to right SAR-Significant-Wave-Height with (**a**) 0.0 m, (**b**) 1.0 m, and (**c**) 2.0 m.

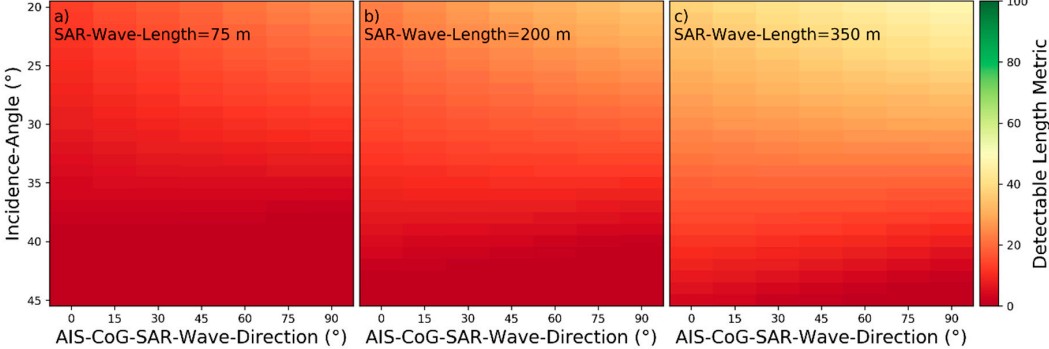

**Figure 30.** Detectability heatmaps for accumulated port and starboard Kelvin wake arms based on AIS-CoG-SAR-Wave-Direction, Incidence-Angle and from left to right SAR-Wave-Length with (**a**) 75 m, (**b**) 200 m, and (**c**) 350 m.

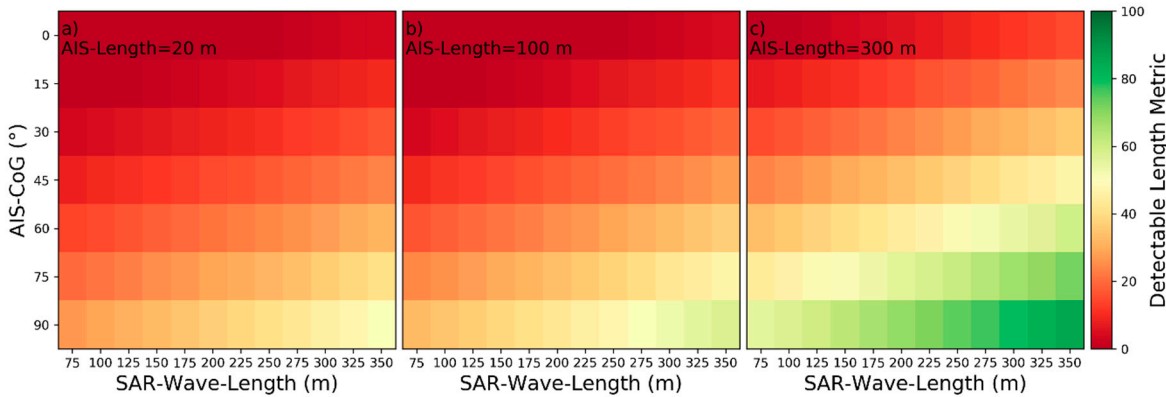

**Figure 31.** Detectability heatmaps for accumulated port and starboard Kelvin wake arms based on SAR-Wave-Length, AIS-CoG and from left to right AIS-Length with (**a**) 20 m, (**b**) 100 m, and (**c**) 300 m.

**Table 6.** Categorization of influencing parameters by characteristics of influences for Kelvin wake arms.

| Characterized Influencing Parameter $x_c$ | Characteristic of Influence | Relevant Figure |
|---|---|---|
| AIS-Vessel-Velocity $(x_1)$ | **Positive independent monotonic influence** $f_w{}'(x_c, X_{o \neq c}) > 0$ | Figure 25, Figure 26, Figure 27 and Figure 28 |
| AIS-Length $(x_2)$ | **Positive independent monotonic influence** $f_w{}'(x_c, X_{o \neq c}) > 0$ | Figure 25 and Figure 31 |
| AIS-CoG $(x_3)$ | **Positive independent monotonic influence** $f_w{}'(x_c, X_{o \neq c}) > 0$ | Figure 25 and Figure 31 |
| Incidence-Angle $(x_4)$ | **Negative independent monotonic influence** $f_w{}'(x_c, X_{o \neq c}) < 0$ | Figure 26, Figure 28 and Figure 30 |
| SAR-Wind-Speed $(x_5)$ | **Negative independent monotonic influence** $f_w{}'(x_c, X_{o \neq c}) < 0$ | Figure 26, Figure 27, Figure 28 and Figure 29 |
| SAR-Significant-Wave-Height $(x_6)$ | Negative independent monotonic influence $f_w{}'(x_c, X_{o \neq c}) < 0,$ but not pronounced, thus **no influence** $f_w{}'(x_c, X_{o \neq c}) \approx 0$ | Figure 29 |
| SAR-Wave-Length $(x_7)$ | **Positive independent monotonic influence** $f_w{}'(x_c, X_{o \neq c}) > 0$ | Figure 27, Figure 30 and Figure 31 |
| AIS-CoG-SAR-Wave-Direction $(x_8)$ | **Interdependent influence** with Incidence-Angle positive monotonic influence for $x_4 \leq 35° f_w{}'(x_c, X_{o \neq c}) < 0$ negative monotonic influence for $x_4 > 35° f_w{}'(x_c, X_{o \neq c}) > 0$ (not pronounced) | Figure 30 |
| AIS-CoG-WRF-Wind-Direction $(x_9)$ | **No influence** $f_w{}'(x_c, X_{o \neq c}) = 0$ | Figure 29 |

**Table 7.** Categorization of Influencing Parameters by Characteristics of Influences for Divergent Waves.

| Characterized Influencing Parameter $x_c$ | Characteristic of Influence | Relevant Figure |
|---|---|---|
| AIS-Vessel-Velocity $(x_1)$ | **Positive independent monotonic influence** $f_w{}'(x_c, X_{o \neq c}) > 0$ | Figure 32, Figure 33 and Figure 34 |
| AIS-Length $(x_2)$ | **One-peaked maximum influence (HH)** $f_w{}'(x_{c,peak}, X_{o \neq c}) = 0$ and $f_w{}''(x_{c,peak}, X_{o \neq c}) < 0$ with $x_{c,peak} \approx 200$ m (neither pronounced, nor robust) **Positive independent monotonic influence (VV)** $f_w{}'(x_c, X_{o \neq c}) > 0$ | Figure 32 |
| AIS-CoG $(x_3)$ | **One-peaked maximum influence** $f_w{}'(x_{c,peak}, X_{o \neq c}) = 0$ and $f_w{}''(x_{c,peak}, X_{o \neq c}) < 0$ with $x_{c,peak} \approx 45°$ (HH) and $x_{c,peak} \approx 30°$ (VV) | Figure 32 |
| Incidence-Angle $(x_4)$ | **Negative independent monotonic influence** $f_w{}'(x_c, X_{o \neq c}) < 0$ | Figure 33 and Figure 34 |
| SAR-Wind-Speed $(x_5)$ | **Negative independent monotonic influence** $f_w{}'(x_c, X_{o \neq c}) < 0$ | Figure 33 and Figure 34 |

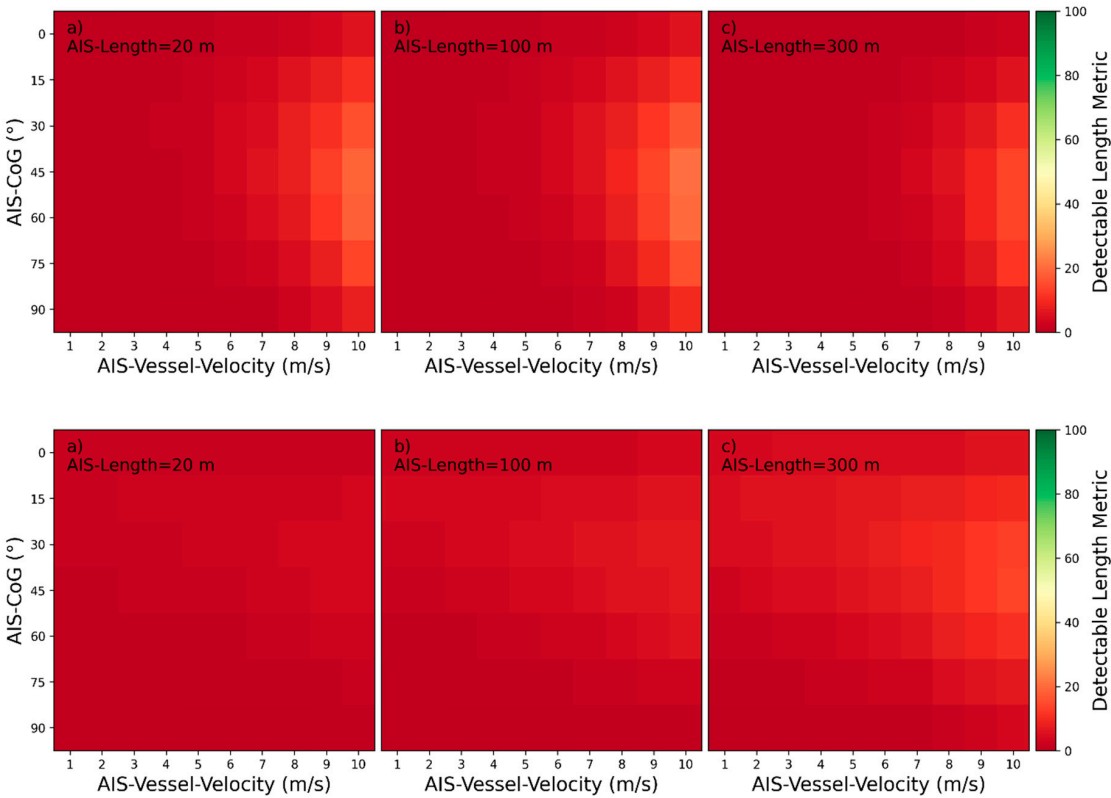

**Figure 32.** Detectability heatmaps for divergent waves based on AIS-Vessel-Velocity, AIS-CoG and from left to right AIS-Length with (**a**) 20 m, (**b**) 100 m, and (**c**) 300 m (**top plots**: HH-polarization, **bottom plots**: VV-polarization).

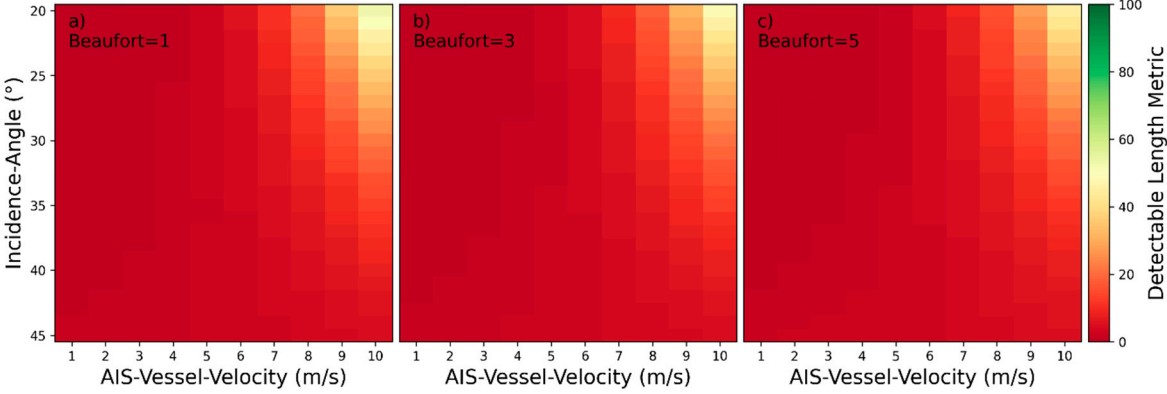

**Figure 33.** Detectability heatmaps for divergent based on AIS-Vessel-Velocity, Incidence-Angle and from left to right Beaufort numbers with (**a**) 1 bft, (**b**) 3 bft, and (**c**) 5 bft.

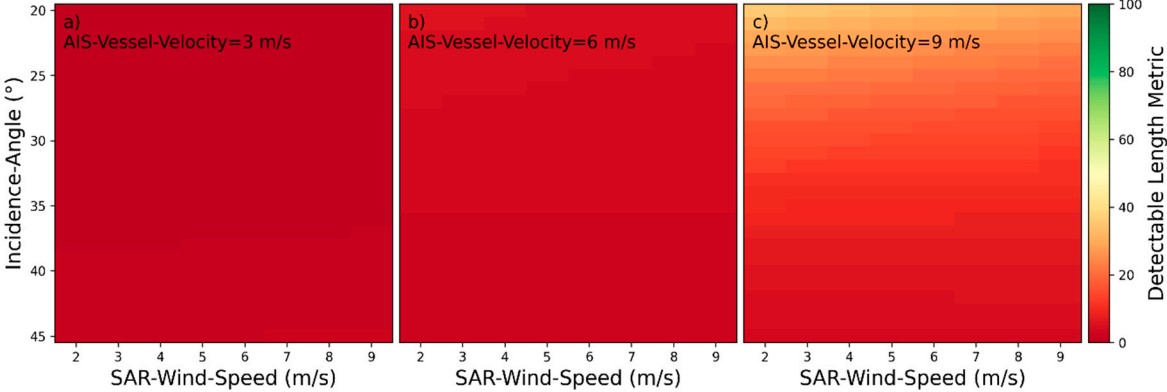

**Figure 34.** Detectability heatmaps for divergent based on SAR-Wind-Speed, Incidence-Angle and from left to right AIS-Vessel-Velocity with (**a**) 3 m/s, (**b**) 6 m/s, and (**c**) 9 m/s.

*3.6. Categorization of Influencing Parameters by Characteristics of Influences for Transverse Waves*

The plots in this subsection are based on the binary SVM classification model instead of the SVR. Therefore, no minimum and the maximum length boundary needs to be defined, as the detectable length metrics are provided by the classifiers directly. For transverse waves, the detectability heatmaps are presented in the Figures 35–37 and based on those heatmaps, the five influencing parameters can be categorized as described in the Table 8.

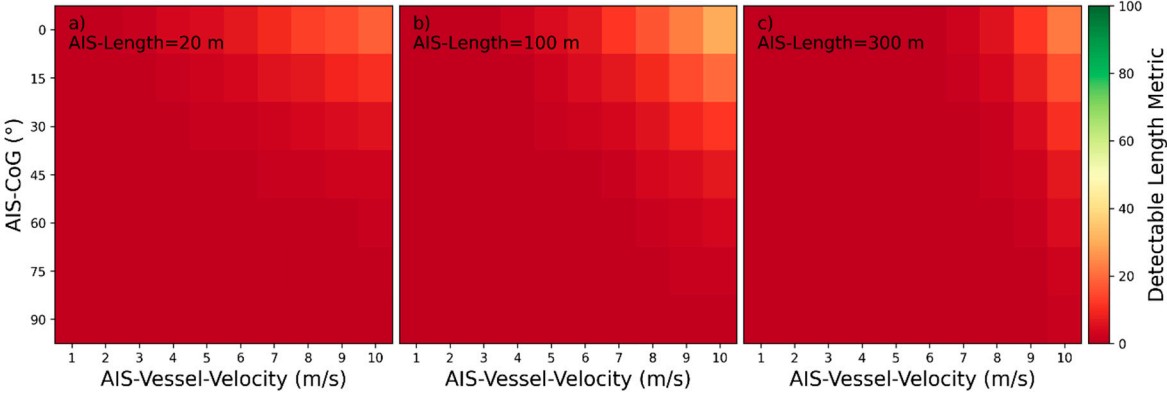

**Figure 35.** Detectability heatmaps for transverse waves based on AIS-Vessel-Velocity, AIS-CoG and from left to right AIS-Length with (**a**) 20 m, (**b**) 100 m, and (**c**) 300 m.

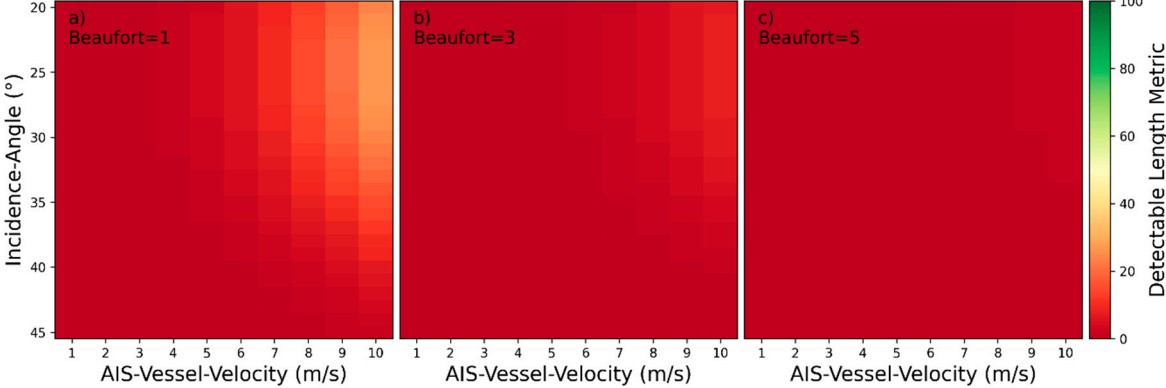

**Figure 36.** Detectability heatmaps for transverse waves based on AIS-Vessel-Velocity, Incidence-Angle and from left to right Beaufort numbers with (**a**) 1 bft, (**b**) 3 bft, and (**c**) 5 bft.

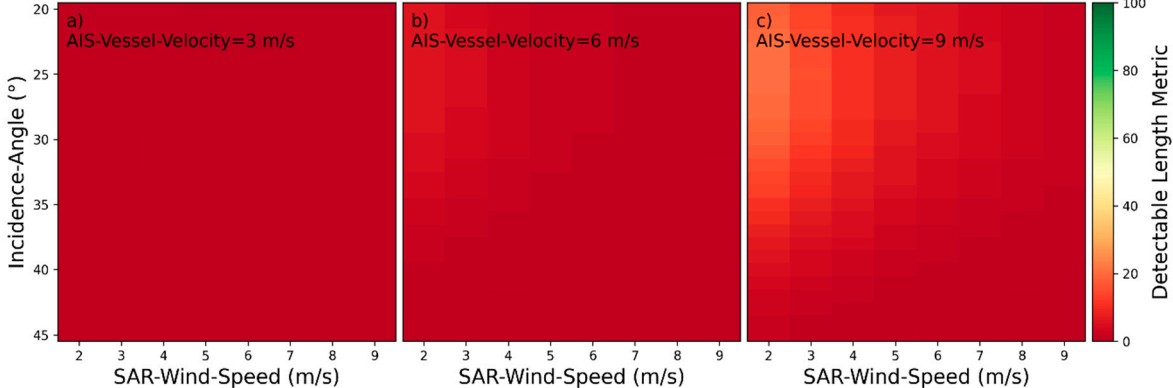

**Figure 37.** Detectability heatmaps for transverse waves based on SAR-Wind-Speed, Incidence-Angle and from left to right AIS-Vessel-Velocity with (**a**) 3 m/s, (**b**) 6 m/s, and (**c**) 9 m/s.

**Table 8.** Categorization of influencing parameters by characteristics of influences for transverse waves.

| Characterized Influencing Parameter $x_c$ | Characteristic of Influence | Relevant Figure |
|---|---|---|
| AIS-Vessel-Velocity ($x_1$) | **Positive independent monotonic influence** $f_w'(x_c, X_{o \neq c}) > 0$ | Figure 35, Figure 36 and Figure 37 |
| AIS-Length ($x_2$) | **Negative independent monotonic influence** $f_w'(x_c, X_{o \neq c}) < 0$ | Figure 35 |
| AIS-CoG ($x_3$) | **Negative independent monotonic influence** $f_w'(x_c, X_{o \neq c}) < 0$ | Figure 35 |
| Incidence-Angle ($x_4$) | One-peaked maximum influence $f_w'\left(x_{c,peak}, X_{o \neq c}\right) = 0$ and $f_w''\left(x_{c,peak}, X_{o \neq c}\right) < 0$ with $x_{c,peak} \approx 25°$, but not robust, thus **negative independent monotonic influence** | Figure 36 and Figure 37 |
| SAR-Wind-Speed ($x_5$) | Negative independent monotonic influence $f_w'(x_c, X_{o \neq c}) < 0$, but not robust, thus **no influence** $f_w'(x_c, X_{o \neq c}) \approx 0$ | Figure 36 and Figure 37 |

*3.7. Categorization of Influencing Parameters by Characteristics of Influences for V-Narrow Wake Arms*

For V-narrow wakes the minimum length boundary is $l_t^{min} = 200$ m and the maximum length boundary is $l_t^{max} = 1250$ m. By the minimum length boundary, the near-hull turbulence around the ship is discarded. The maximum length boundary for the heatmap plots in the Figures 38–44 roughly corresponds to the length of the port and starboard V-narrow wake arms at the 80th percentile over all wake samples with measured V-narrow wake arms. According to the plots, the nine influencing parameters can be categorized as described in the Table 9.

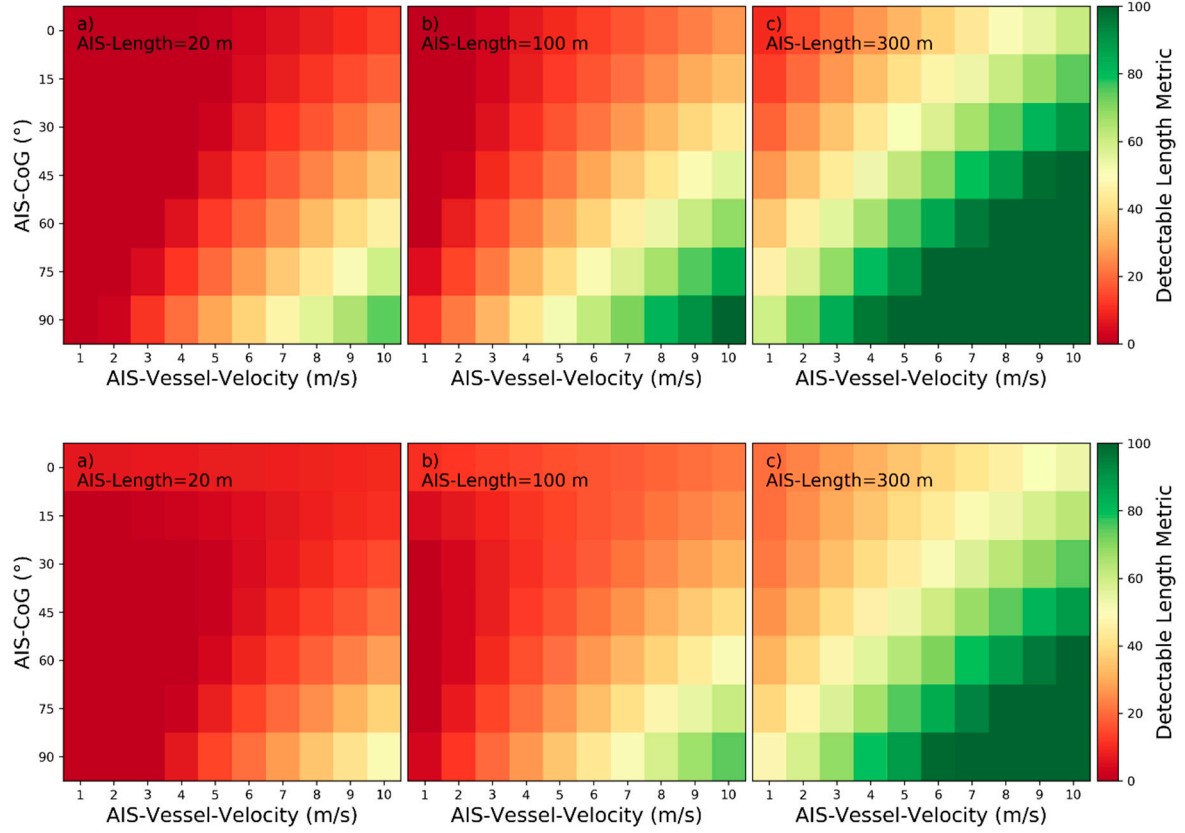

**Figure 38.** Detectability heatmaps for accumulated port and starboard V-narrow wake arms based on AIS-Vessel-Velocity, AIS-CoG and from left to right AIS-Length with (**a**) 20 m, (**b**) 100 m, and (**c**) 300 m (**top plots**: HH-polarization, **bottom plots**: VV-polarization).

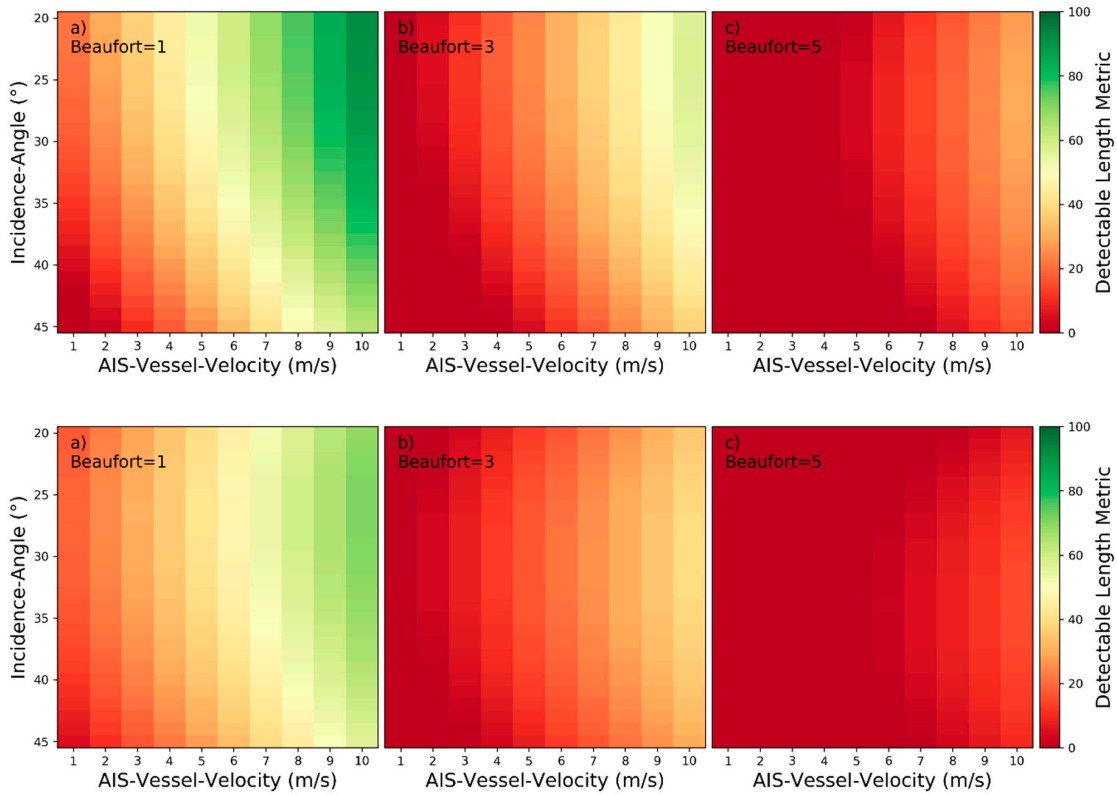

**Figure 39.** Detectability heatmaps for accumulated port and starboard V-narrow wake arms based on AIS-Vessel-Velocity, Incidence-Angle and from left to right Beaufort numbers with (**a**) 1 bft, (**b**) 3 bft, and (**c**) 5 bft (**top plots**: HH-polarization, **bottom plots**: VV-polarization).

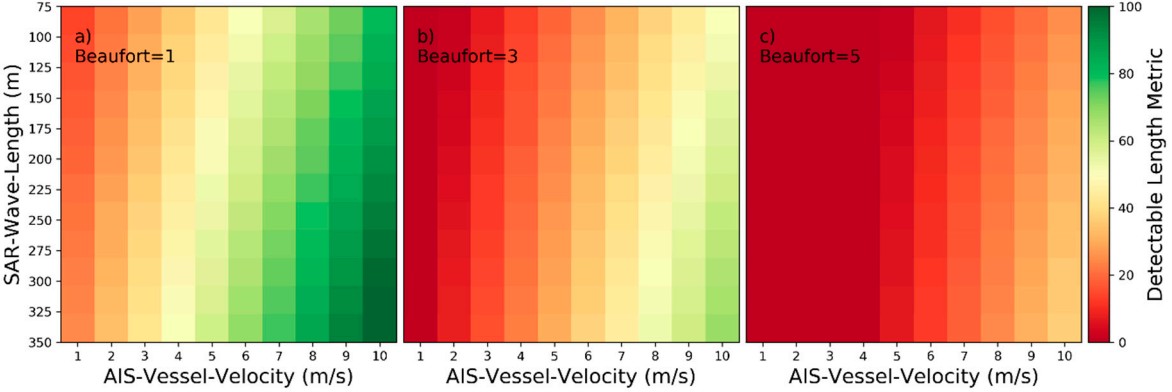

**Figure 40.** Detectability heatmaps for accumulated port and starboard V-narrow wake arms based on AIS-Vessel-Velocity, SAR-Wave-Length and from left to right Beaufort numbers with (**a**) 1 bft, (**b**) 3 bft, and (**c**) 5 bft.

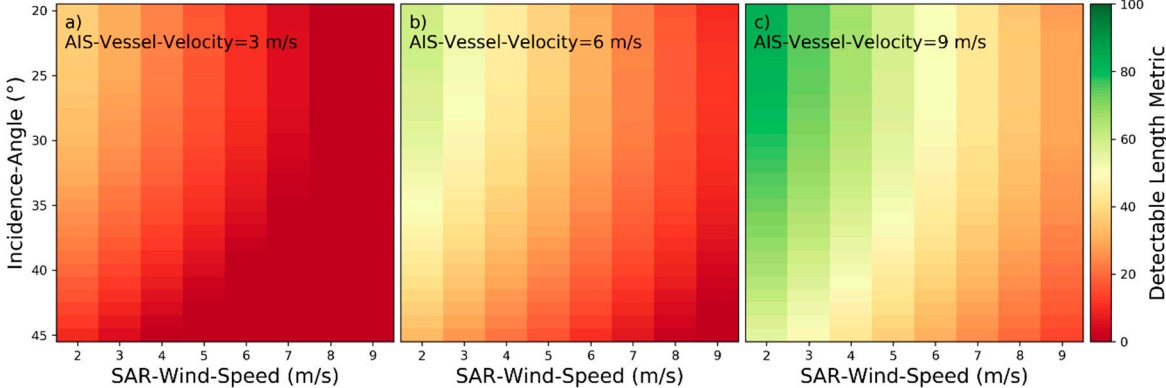

**Figure 41.** Detectability heatmaps for accumulated port and starboard V-narrow wake arms based on SAR-Wind-Speed, Incidence-Angle and from left to right AIS-Vessel-Velocity with (**a**) 3 m/s, (**b**) 6 m/s, and (**c**) 9 m/s.

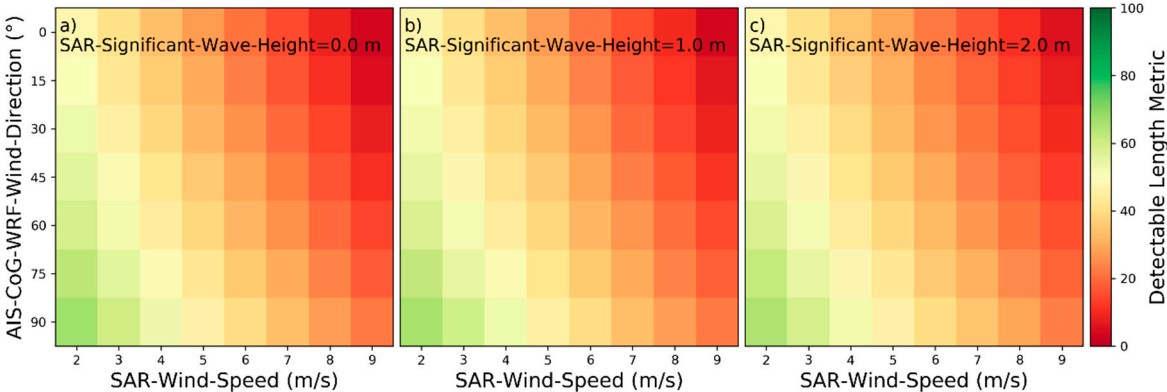

**Figure 42.** Detectability heatmaps for accumulated port and starboard V-narrow wake arms based on SAR-Wind-Speed, AIS-CoG-WRF-Wind-Direction and from left to right SAR-Significant-Wave-Height with (**a**) 0.0 m, (**b**) 1.0 m, and (**c**) 2.0 m.

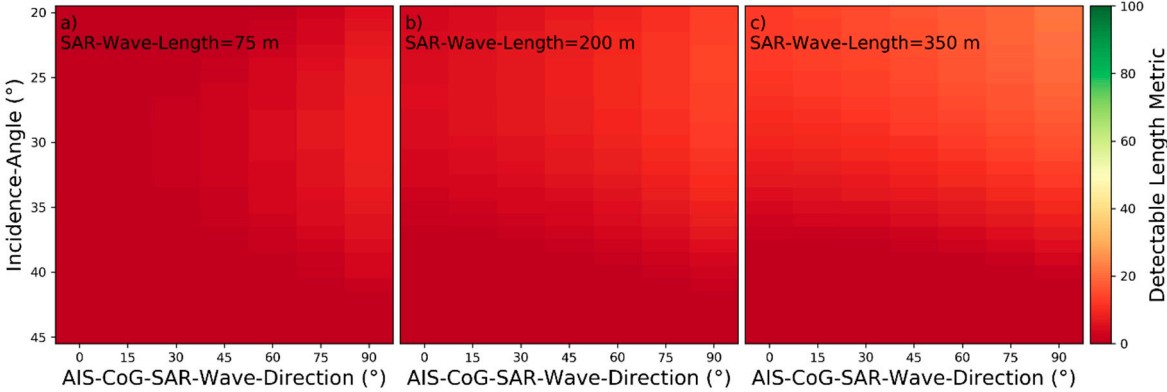

**Figure 43.** Detectability heatmaps for port V-narrow wake arms based on AIS-CoG-SAR-Wave-Direction, Incidence-Angle and from left to right SAR-Wave-Length with (**a**) 75 m, (**b**) 200 m, and (**c**) 350 m.

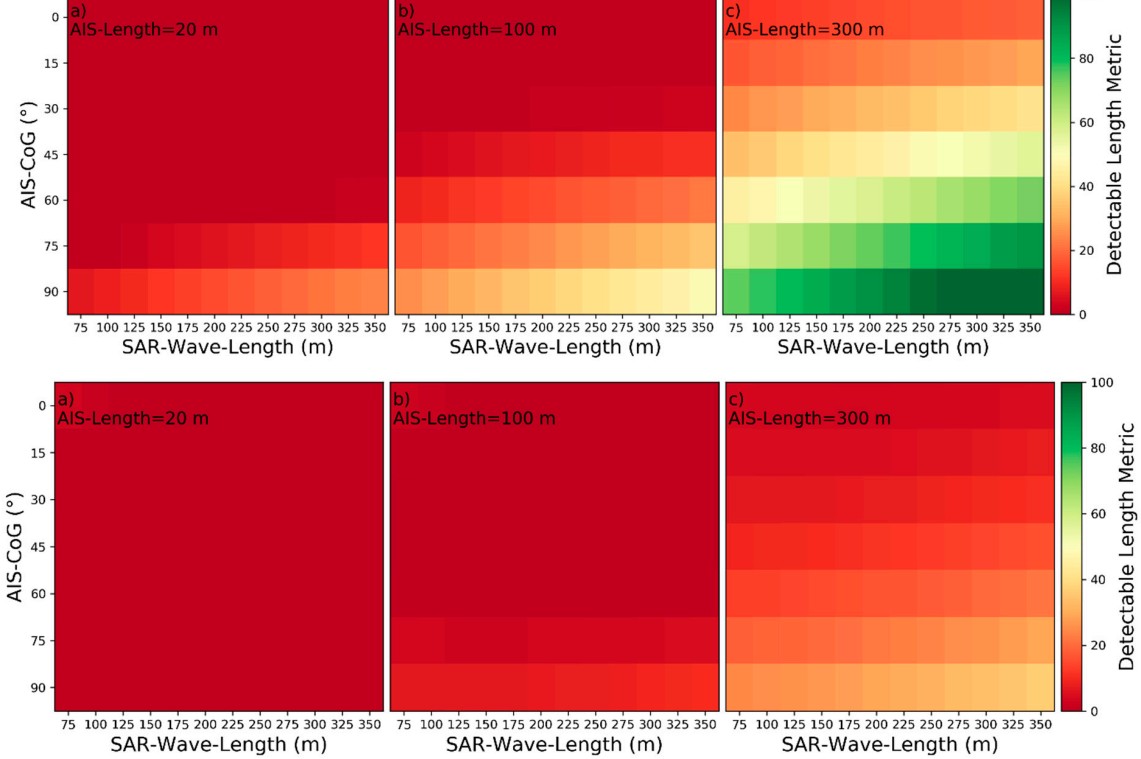

**Figure 44.** Detectability heatmaps for port V-narrow wake arms based on SAR-Wave-Length, AIS-CoG and from left to right AIS-Length with (**a**) 20 m, (**b**) 100 m, and (**c**) 300 m (**top plots**: HH-polarization, **bottom plots**: VV-polarization).

**Table 9.** Categorization of influencing parameters by characteristics of influences for V-narrow wake arms.

| Characterized Influencing Parameter $x_c$ | Characteristic of Influence | Relevant Figure |
|---|---|---|
| AIS-Vessel-Velocity $(x_1)$ | **Positive independent monotonic influence** $f_w{}'\left(x_c, X_{0 \neq c}\right) > 0$ | Figure 38, Figure 39, Figure 40 and Figure 41 |
| AIS-Length $(x_2)$ | **Positive independent monotonic influence** $f_w{}'\left(x_c, X_{0 \neq c}\right) > 0$ | Figure 38 and Figure 44 |
| AIS-CoG $(x_3)$ | **Positive independent monotonic influence (HH)** $f_w{}'\left(x_c, X_{0 \neq c}\right) > 0$ Interdependent influence (VV) with AIS-Vessel-Velocity, AIS-Length and SAR-Wave-Length one-peaked minimum influence for $x_1 \leq 8$ m/s, $x_2 \leq 100$ m and $x_7 \leq 175$ m $f_w{}'\left(x_{c,peak}, X_{0 \neq c}\right) = 0$ and $f_w{}''\left(x_{c,peak}, X_{0 \neq c}\right) > 0$ with $x_{c,peak} \approx [30°, 45°]$ (not robust) positive monotonic influence for $x_1 > 9\frac{m}{s}$, $x_2 > 100$ m and $x_7 > 150$ m $f_w{}'\left(x_c, X_{0 \neq c}\right) > 0$ Due to robustness check **positive independent monotonic influence** is assumed also for **VV** | Figure 38 and Figure 44 |
| Incidence-Angle $(x_4)$ | **Negative independent monotonic influence (HH)** $f_w{}'\left(x_c, X_{0 \neq c}\right) < 0$ One peaked maximum influence (VV) $f_w{}'\left(x_{c,peak}, X_{0 \neq c}\right) = 0$ and $f_w{}''\left(x_{c,peak}, X_{0 \neq c}\right) > 0$ with $x_{c,peak} \approx [25°, 35°]$ (not robust) Due to robustness check **negative independent monotonic influence** is assumed also for **VV** | Figure 39 and Figure 41 |
| SAR-Wind-Speed $(x_5)$ | **Negative independent monotonic influence** $f_w{}'\left(x_c, X_{0 \neq c}\right) < 0$ | Figure 39, Figure 40, Figure 41 and Figure 42 |
| SAR-Significant-Wave-Height $(x_6)$ | **No influence** $f_w{}'\left(x_c, X_{0 \neq c}\right) = 0$ | Figure 42 |
| SAR-Wave-Length $(x_7)$ | **Positive independent monotonic influence** | Figure 40 and Figure 44 |
| AIS-CoG-SAR-Wave-Direction $(x_8)$ | Contradiction between port, starboard and robustness models, thus **no influence** $f_w{}'\left(x_c, X_{0 \neq c}\right) \approx 0$ | Figure 43 |
| AIS-CoG-WRF-Wind-Direction $(x_9)$ | **Positive independent monotonic influence** $f_w{}'\left(x_c, X_{0 \neq c}\right) < 0$ | Figure 42 |

### 3.8. Categorization of Influencing Parameters by Characteristics of Influences for Ship-Generated Internal Waves

As specified in Table 1, only for around 2% of all wake samples are ship-generated internal waves are detectable. A statistical analysis is not possible with this low amount of data. The only definite result is that ship-generated internal waves are detectable on TerraSAR-X high resolution imagery, but only in rare cases.

The fact that the few ship-generated internal waves were found with strongly varying influencing parameters, supports the statement from other researchers that internal waves' occurrence is mainly based on the stratification of ocean layers [18,19,23,24], but not on image acquisition settings, environmental conditions or ship properties.

## 4. Discussion

Most of the results presented in the previous Section 3 are in agreement with the results published by other researchers on the detectability of wake components. Nevertheless, the method presented here for modelling the wake component's detectability on the basis of real data together with SVMs now provides the opportunity to confirm those statements, and to fill in missing statements.

Many of the previous studies on wakes referenced here do not explicitly discuss the term "detectability". Often the impacts of the influencing parameters on physical

characteristics, e.g., dampening ratios, wave amplitudes, normalized radar cross section, etc., are described. Therefore, such previous studies had to be interpreted in terms of wake component detectability, i.e., no influence, positive/negative independent monotonic influence, one-peaked minimum/maximum influence and interdependent influence. The introduced characteristics of influences on wake component detectability were not used by previous researchers themselves.

For wake components with starboard and port elements, the results in Sections 3.4 and 3.7 are based on accumulated wake component lengths from the respective datasets containing either port or starboard elements. Nevertheless, for robustness checking the two models of one wake component for separate port and starboard elements were also trained, respectively. The resulting heatmaps for both elements are not identical. Though, in theory, for a model based on a dataset with infinite size, they would be identical, because the elements are axially symmetric to the ship track and the AIS-CoG has been projected to the $0°$ to $90°$ value range, which would project the starboard and port elements on top of each other. However, a qualitative comparison of heatmaps for starboard and port elements shows consistency. As the models are based on a finite amount of real data and additionally are affected by human error introduced during the manual inspection procedure, this consistency indicates that the applied modelling method is robust.

By changing the SVM hyperparameters to non-optimal settings in terms of 10-fold cross validated accuracy, the robustness of models was investigated further. Comparisons of heatmaps from multiple such models showed that most hyperparameters do not change the characteristics of influences when altered. However, increasing the Gamma-hyperparameter to values above zero has an effect on the characteristics: In some cases, one-peaked maxima or minima become flattened, which decreases the magnitude of influence or even results in no influence. Additionally, some independent monotonic influences decrease in magnitude. This behavior can be explained by the fact that an increase in the Gamma-hyperparameter reduces the allowed curvature of the model's hyperplanes. Therefore, this robustness check is in total considered as passed, but the respective characteristics with discrepancies are marked, when the results are not robust.

The different settings of SVM hyperparameters were combined with respectively two reduced versions of each wake component dataset, as introduced at the end of Section 2.3. The majority of characteristics remain unchanged. As both filtering methods are based on removing samples with parameter values below or above the 10th or 90th percentile, mostly samples with rare features are removed. As a result, some independent monotonic influences, some one-peaked maxima influences as well as some interdependent influences are reduced in magnitude or even become characterizable as having no influence. As a reduction in magnitude of influence is not considered as change of characteristic of influence, these results of robustness check are considered as passed. In the case of discrepancies, the respective characteristics are marked as being non-robust.

In general, consistency of heatmaps can be observed in the results between HH-polarization and VV-polarization. That the difference in polarization does not considerably affect the characteristics of influence on wake component's detectability is also supported by other studies, e.g., in [9,11,27]. Nevertheless, some inconsistencies between HH- and VV-polarization exist, but in terms of characteristics of influence these inconsistencies are mostly negligible.

Generally, non-robust characteristics mainly occur when the influence is also not pronounced. This means that most non-robust characteristics actually imply that the respective influencing parameter has no influence. In the following subsections, the impact of each influencing parameter on the detectability of the investigated wake components is explained. All non-robust characteristics or characteristics showing relevant inconsistencies between HH- and VV-polarization are discussed.

It should be noted again that the length of the individual wake components is used here as indicator for their detectability. This means the $DLM_w$ represents an expected detectable wake component length under certain conditions and should not be confused

with probability of detection. In contrast, the binary classification model used for detectability analysis of transverse and divergent waves really provides a probability, but this probability is also just an indication for the probability of detection.

### 4.1. AIS-Vessel-Velocity

For all wake components, AIS-Vessel-Velocity shows positive independent monotonic influence. With higher vessel velocity, the ocean surface area affected by the ship's movement is larger. Therefore, it is straightforward to conclude that all wake components with $DLM_w$ indicated by wake component length are more readily detectable, i.e., near-hull turbulences, turbulent wakes, Kelvin wake arms and V-narrow wake arms.

The energy transferred to the water by ship's movement is also higher at higher vessel velocities, which result in more water mass being pushed away. Thus, the Kelvin wake's divergent waves have higher amplitude, as already described in Section 1.2.2 or by [21,22], which implies better detectability

According to [21,22], larger Froude numbers, coming with faster ship's movements, imply lower amplitudes of transvers waves. However, our model estimates that also the transverse waves are also more detectable with higher AIS-Vessel-Velocity (see Figures 35 and 36). This can be explained by the fact that with larger Froude numbers the wavelength of the transverse waves increases [21]. While wave systems with long wavelength can be better imaged by SAR, the individual waves in wave systems with short wavelength cannot be resolved anymore. The so-called cut-off effect is responsible, which in turn can be justified by the velocity bunching mechanism [10,30]. Therefore, only transverse waves with wavelength above a certain threshold can be detected, which is reproduced in the detectability model by positive independent monotonic influence.

### 4.2. AIS-Length

For all lineated wake components, the presented results show higher $DLM_w$ for longer vessels, i.e., positive independent monotonic influence for AIS-Length. It can be assumed that longer vessels also have a deeper draft and are more likely displacement ships than planing ships. Therefore, the vessel's mass increases cubically rather than linearly with the vessel's length in most cases. This means that the energy transferred to the water by moving vessels is considerably higher for larger vessels, which results in stronger turbulences and higher amplitudes of the wave systems produced. In addition, the higher draft of displacement ships introduces more air into deeper water layers resulting in longer ascending times of bubbles and surfactants. Therefore, larger vessel sizes increase the detectability of near-hull turbulences, turbulent wakes, V-narrow wake arms and Kelvin wake arms.

For divergent waves the model reproduces positive independent monotonic influence for VV-polarization and one-peaked maximum influence at high AIS-Length of 200 m for HH-polarization. This result is in contradiction to [21,22], where it was stated that smaller Froude numbers, as a result of longer ship lengths (see Equation (1)), mean lower divergent wave amplitudes. The lower divergent wave amplitudes actually imply a lower detectability of the divergent waves with increasing AIS-Length. On the other hand, the wavelength of divergent waves increases for larger ship lengths [21]. Due to the cut-off effect, these longer wavelengths are better detectable than shorter wavelengths [30]. Thus, the positive monotonic influence independently for VV-polarization and until the one-peaked maximum for HH-polarization (as presented in Figure 32) is rather not pronounced and almost no influence of AIS-Length on the detectability of divergent waves can be assumed. It seems in terms of detectability, the increase in divergent wave's wavelength coming with larger vessels and lower Froude numbers almost compensates the decrease in divergent wave's amplitudes. This explains the one-peaked maximum observable for HH-polarization: When wavelengths of divergent waves are getting long enough, the cut-off effect becomes irrelevant and the detectability decreases due to smaller wave's amplitudes. The drop of $DLM_w$ after the peak at AIS-Length of 200 m is pronounced

in the heatmaps, but it should be noted that the one-peaked maximum is not robustly reproduced by all models. Models trained with higher Gamma-hyperparameter also reproduce positive independent monotonic influence for HH-polarization. However, as the one-peaked maximum perfectly makes sense due to the above explanation, it should also be present for VV-polarization and the failed robustness check for HH-polarization is negligible.

In contrast, transverse waves have higher wave amplitudes and shorter wavelengths with increasing ship lengths and concurrently decreasing Froude numbers. The detectability model reveals negative independent monotonic influence (see Figure 35). This means, smaller vessels are better for the detectability of transverse waves, because they produce transverse waves with longer wavelengths, which are better detectable in SAR as already explained in the previous paragraph.

Although positive independent monotonic influence is shown in the results for Kelvin wake arms, the increase in $DLM_w$ from 5 m to 350 m AIS-Length is not pronounced so that actually no influence can be assumed. The Kelvin wake arms consist of cusp waves, which are a product of constructive interference between transverse and divergent waves. That an increase in AIS-Length does not considerably increase the $DLM_w$ of Kelvin wake arms implies that with the respectively decreasing Froude numbers coming with the increase in AIS-Length the decreasing transverse wave's amplitude is compensated by the increasing divergent wave's amplitudes. Thus, the divergent wave's amplitudes must grow much faster than the transverse wave's amplitudes shrink.

*4.3. AIS-CoG*

The relative movement direction of the vessel with respect to the radar beam's looking direction, i.e., AIS-CoG, is the influencing parameter with most varying characteristics of influences on wake component detectability. The statements from literature summarized in Section 1.2 are all in agreement with each other by specifying that azimuthal ship movement is best for the detection of most wake components, which means positive independent monotonic influence. In this subsection is discussed that the presented detectability models show partially opposed results.

According to the results, for near hull turbulences the influence of AIS-CoG is interdependent on AIS-Vessel-Velocity. For slow vessels with AIS-Vessel-Velocity below or equal 6 m/s, AIS-CoG has no influence on the detectability. Only for faster vessels above this threshold AIS-CoG has negative monotonic influence, which means near-hull turbulences are better detectable when created by ships with movement direction parallel to range. Fast vessels, especially planing ships, often drag behind one high amplitude wave with static distance to their sterns, which then resolves into port and starboard divergent waves. This wave provides better backscatter conditions when the radar signal is reflected perpendicularly instead of from the either side. However, the robustness check failed and the characterization together with the above explanation cannot be definitely postulated here.

The presented detectability model reproduces different characteristics for detectability of turbulent wakes. According to Gade et al. [26] the polarization has no effect on the damping ratio of surface films. Therefore, the inconsistency between HH- and VV-polarization is unexpected, and it can be concluded that the research on detectability of surface films is not a perfectly accurate supplement for the missing statements on the detectability of turbulent wakes. However, no change in the overall detectability between HH- and VV-polarization is indicated by the detectability model, which supports the postulation of Gade at al. In detail, the model reproduces interdependent influence with SAR-Wave-Length for HH-polarization and negative independent monotonic influence for VV-polarization. For HH-polarization, until SAR-Wave-Length of 200 m turbulent wakes are better detectable, when the vessels move neither parallel to range nor parallel to azimuth, but this characteristic is not robustly reproduced by other models trained for checking robustness. Some models reproduce no influence until SAR-Wave-Length of 200 m and other models reproduce positive independent monotonic influence without interdependency to sea state's

wavelength. In conclusion, with larger sea state's wavelengths and HH-polarization, ships moving parallel to azimuth direction are definitely more detectable than ships moving parallel to range direction. In fact, positive independent monotonic influence was assumed generally by Lyden et al., but this assumption is not supported by justification [8]. It should also be noted that Lyden at al. were only able to recognize sea state with long wavelengths with their used low-resolution sensors techniques. For VV-polarization no robust characterization can be postulated, as robustness checking failed entirely: some models reproduce positive and some models negative independent monotonic influence.

Positive independent monotonic influence of AIS-CoG is reproduced by the detectability models for Kelvin wake arms. This means that the Kelvin wake arms together with the cusp waves on the arms are oriented perpendicularly to the radar beam's looking direction, which results in increased Bragg scattering. During manual inspection was recognized that even in cases where the ship's CoG is perfectly parallel to azimuth direction, mostly one Kelvin wake arm is brighter than the other. This observation supports the statement from Hennings et al. [9] that the traveling direction of cusp waves towards to radar beam's looking direction would imply higher backscatter compared to cusp waves traveling away from the sensor.

For divergent waves, a one-peaked maximum influence is reproduced by the detectability model. According to the presented model, highest detectability is achieved when the ship's CoG relative to the range direction is around 45° for HH-polarization and 30° for VV-polarization. The propagation direction of divergent waves relative to AIS-CoG lies within the interval [35°16′, 90°] [9]. Only in close vicinity to the ship's hull the divergent waves have a relative angle of 90° to the ship track, but this region is dominated by near-hull turbulences. Additionally, the wavelength of divergent waves is too small to be sufficiently distinctive for the sensor with relative angles close to 90°. The majority of divergent waves are detected outside the Kelvin envelope. This means, that the divergent waves detectable on SAR rather have a relative direction of 35°16′ than 90°. When the ship's movement direction is neither parallel to range nor parallel to azimuth, this implies that major parts of port or starboard divergent waves are moving roughly parallel to range direction. This movement direction makes the single wave crests and troughs better discriminable due to higher tilt and hydrodynamic modulation [30]. During the robustness check the one-peak maximum was frequently reproduced with variation around 45°. Only for higher Gamma-hyperparameters the one-peaked got flattened, but the postulated characteristics are still considered robust.

Negative independent monotonic influence is indicated by the model for transverse waves. According to Alpers et al. [30], the tilt and hydrodynamic modulation of ocean waves is higher for HH-polarization and propagation direction parallel to range. So, it can be assumed that the negative independent monotonic influence is reproduced by the model, because the higher tilt and hydrodynamic modulation makes the wave crests and troughs of transverse waves better discriminable and therefore better detectable. This result is in contradiction to assumptions by Lyden at al. [8], who expected positive independent monotonic influence.

In [8], [14] or [32] for V-narrow wake arms the interdependency to other ship properties and sea state's wavelength is not taken into account and only positive independent monotonic influence is assumed. The heatmaps for HH-polarization confirm this assumption. However, the presented detectability models for VV-polarization reveal that AIS-CoG has an interdependent influence on the detectability of V-narrow wake arms. The assumed positive monotonic influence is only recognizable with AIS-Vessel-Velocity above 8 m/s, AIS-Length above 100 m or SAR-Wave-Length above 175 m. With interdependent parameter values below these settings, mainly a one-peaked minimum can be derived from the heatmaps (see Figures 38 and 44), which is located in the interval of $x_{c,peak} \approx 30°$ to $x_{c,peak} \approx 45°$. Thus, if the interdependent parameters have settings with low values, V-narrow wakes are better detectable, when the ships are moving either parallel to range or parallel to azimuth, but not diagonally to both directions. By modelling the detectability

for V-narrow wakes under HH-polarization separately for the port and starboard side, positive independent monotonic influence is only reproduced by the model for the starboard V-narrow wake arm. The model for the port V-narrow wake arm shows the same characteristics as indicated for VV-polarization. However, the one peaked minimum influence is for both interdependent influences of AIS-CoG less pronounced than the positive monotonic influence, which is a result of the basically positive independent monotonic influence of the three interdependent parameters. It should be noted that both V-narrow wake arms are rarely detectable, when the vessel is moving parallel to the range direction, while frequently both V-narrow wakes arms are detectable aft ships with movement parallel to azimuth. As V-narrow wakes are only detectable due to Bragg scattering, for ships with movement parallel to range, the Bragg criterion seems only be satisfiable for either port or starboard wavelengths circularly created by the ships' hull. The robustness check finally reveals that all other models with higher Gamma-hyperparameter and reduced training datasets reproduce positive independent monotonic influence for both V-narrow wake arms as well as for both polarizations. Therefore, that V-narrow wakes are better detectable when ships are moving parallel to azimuth, as assumed in [8,14,32], can basically be confirmed here.

Finally, an assumption can also be provided here for ship-generated internal waves. All internal wave samples detected in the dataset were caused by vessel's moving with AIS-CoG above 45°. It can therefore be concluded that vessel's movement parallel to azimuth is better for the detection of ship-generated internal waves. This effect has not yet been described in the literature and was observed for the first time in the scope of this work.

### 4.4. Incidence-Angle

Negative independent monotonic influence can be recognized for all wake components, except near-hull turbulences and transverse waves. Negative independent monotonic influence of all other wake components was stated by [8,9,25] and can herby be confirmed. For V-narrow wakes under VV-polarization the model actually reproduces a one-peaked maximum influence, but all other models trained for robustness checking show negative independent monotonic influence similar to V-narrow wakes under HH-polarization. Therefore, negative independent monotonic influence is also generally assumed for V-narrow wake arms.

Interdependent influence with AIS-Vessel-Velocity is reproduced by the model for near-hull turbulences. For slow vessels with $x_{AIS-Vessel-Velocity} \leq 3 \, \text{m/s}$, almost no influence of the incidence angle is indicated. For faster vessels, the influence is positive monotonic. Vessels with notable movement through the water create high amplitude waves and wave breaking in their close vicinity. Those waves are dominating the imaged ocean backscatter, as they provide high backscattering conditions [48]. Therefore, the near-hull turbulence is better detectable with increasing incidence angle. In [12] it is stated that Incidence-Angle has an interdependent influence to AIS-Vessel-Velocity. This can now be explained by the fact that for unfavorable wake detection conditions the wakes of fast vessels were detected only because of the near-hull turbulences.

For transverse waves a one-peaked maximum is reproduced by the model, which is with $x_{Incidence-Angle,peak} \approx 25°$ so close to the lower limit of the considered value range of Incidence-Angle ($[20°, 45°]$) that negative independent monotonic influence is assumed. Negative independent monotonic influence is also reproduced by some models trained with increased Gamma-hyperparameter or reduced samples size of the training data. Due to lower tilt and hydrodynamic modulation for higher incidence angles, the distinctiveness of wave crests and troughs decreases [30]. Therefore, lower incidence angles are better suited for detecting transverse waves.

### 4.5. SAR-Wind-Speed

SAR-Wind-Speed is derived from SAR by the evaluation of mean radar backscatter reflected from the ocean surface to the satellite [39]. Within the considered value range

from 2 m/s to 10 m/s the mean radar backscatter grows proportionally. With the increased amount of radar backscatter the level of speckle noise also increases, making any wake component less distinguishable from the ocean background. Therefore, a negative independent monotonic influence is visible in all heatmaps and holds for all wake components. However, for transverse waves the robustness check failed: Some models reproduced one-peaked maxima and some even slight positive independent monotonic influences. In conclusion, the effect of SAR-Wind-Speed on the detectability of transverse waves is assumed negligible. This can be explained by the fact that most transverse waves have high wavelengths, when they are imaged by SAR and are not affected by the cut-off effect. Tilt and hydrodynamic modulation are responsible for their appearance. Those modulations required a minimum wind speed being present, but in the value range of 2 m/s to 10 m/s are then hardly affected by wind speed.

It should be noted that V-narrow wakes are frequently detectable at wind speeds above 3 m/s in the TS-X dataset used. Thus, the assumption of Lyden et al. [8] that V-narrow wakes would not appear in X-Band SAR imagery cannot be confirmed here.

A statement can be provided for ship-generated internal waves, although detectability has not been modelled and hence no heatmaps are available: Single internal wave samples were found with low and high values of all influencing parameters within the parameter's value ranges, except for SAR-Wind-Speed and AIS-CoG. All internal wave samples were detected under SAR-Wind-Speed below 6 m/s. It can therefore be concluded that internal waves are also better detectable under low wind speed conditions.

### 4.6. SAR-Significant-Wave-Height

The influencing parameter SAR-Significant-Wave-Height correlates with SAR-Wind-Speed, at least for wind sea conditions [28]. Our results show that in comparison to SAR-Wind-Speed for SAR-Significant-Wave-Height no similar characteristics of influences on wake component detectability exist. In almost all models, the SAR-Significant-Wave-Height parameter seems rather irrelevant, although a negative independent monotonic influence was stated by many researchers for different wake components [10,18,29]. Indeed, negative independent monotonic influence is reproduced by most detectability models, but the variation of detectable length metric between 0 m and 3 m SAR-Significant-Wave-Height is not pronounced. Thus, this parameter is characterized as having almost no influence for all wake components. It can be concluded that, in case of swell wave conditions without wind, the signatures of wake components are superimposed on the sea state patterns. For all other sea state conditions, the present wind speed is more relevant for the detectability.

### 4.7. SAR-Wave-Length

While the influence of significant wave height on wake component detectability has been considered by other researchers for turbulent wakes, Kelvin wakes and V-narrow wakes (see Section 1.2), further important sea state parameters have not been taken into account [10,18,29]. Therefore, the presented detectability models are used in this subsection and Section 4.8 to derive new relationships between the sea state parameters and wake component detectability. Due to the high noise of the SAR-Wave-Length parameter, the results stated in this subsection should only be used with caution.

For near-hull turbulences, a vague one-peaked maximum influence is reproduced by the model, but other models with different hyperparameter settings and reduced training datasets show vague positive independent monotonic influence. As the influence is neither pronounced nor robust no influence of SAR-Wave-Length is assumed. White capping in the vicinity of the ship's hull and the created rough whitewater produces significantly higher backscatter compared to most sea state conditions. Therefore, the detectability of near-hull turbulences is hardy affected by the sea state's wavelength.

Positive independent monotonic influence is shown in the results for V-narrow wakes, turbulent wakes and Kelvin wake arms. With longer wavelengths of the surrounding sea state the V-narrow wakes', turbulent wakes' and Kelvin wake arm's detectability is

increased. Shorter wavelengths imply higher ambient ocean backscatter, as they are not resolved as pattern of single waves anymore, especially when they fall below the radar's cutoff wavelength [30]. Similar to the influence of SAR-Wind-Speed on the detectability of wake components, higher ambient ocean backscatter impedes the detectability of wake components. In case of longer sea state wavelengths, the signatures of V-narrow wakes, turbulent wakes and Kelvin wake arms simply superimpose onto the sea state's wave crests and troughs.

*4.8. AIS-CoG-SAR-Wave-Direction*

In addition, AIS-CoG-SAR-Wave-Direction is a parameter considered rarely in terms of the detectability of wake components and also suffers from increased noise, as already stated in the previous Section 4.7. The following assumptions are new and do not take the increased noise level of this parameter into account. Nevertheless, they are in good agreement with theoretical considerations about the physics of SAR imaging of the ocean surface.

The detectability of near-hull turbulences and turbulent wakes is influenced by AIS-CoG-SAR-Wave-Direction interdependently with SAR-Wave-Length. For near-hull turbulences the influence of AIS-CoG-SAR-Wave-Direction is additionally interdependent on Incidence-Angle. It should be noted that for both wake components the influence of sea state' wave direction is neither pronounced nor robust and actually no influence could be assumed. Nevertheless, the slightly indicated influence can be explained as follows: With shorter wavelengths of the sea state, near-hull turbulences and turbulent wakes are better detectable, when the propagation direction of sea state waves is perpendicular to the ship's movement direction. For longer sea state wavelengths AIS-CoG-SAR-Wave-Direction has no influence on the detectability of both wake components as their signatures are simply superimposed onto the sea state pattern. While these characteristics of influence hold generally for turbulent wakes, for near-hull turbulences the incidence angle also has an effect: when the incidence angle is above $x_{Incidence-Angle} > 30°$, AIS-CoG-SAR-Wave-Direction has no considerable influence on the detectability of near-hull turbulences anymore. The reason is that the contrast between the turbulences' white capping and the ocean background increases for higher incidence angles, as already explained in Section 4.4, while the distinctiveness of sea state's wave crests and troughs decreases [30] implying less influence of sea state on the detectability.

According to the detectability model for Kelvin wake arms, AIS-CoG-SAR-Wave-Direction has interdependent influence on the detectability with the Incidence-Angle parameter. For low incidence angles below 35° positive monotonic influence can be observed. For $x_{AIS\text{-}CoG\text{-}SAR\text{-}Wave\text{-}Direction} = 90°$ the sea state's wave crests and troughs are parallel to the ship track, while propagating towards one Kelvin wake arm and away from the other. Whenever the sea state's waves collide with the cusp waves in the Kelvin wake arms, increased white capping is the result. Therefore, the already significantly higher backscatter of the Kelvin wake arms in comparison to the ambient ocean backscatter is even more increased. With sea state's propagation direction perpendicular to the ship's movement direction, these collisions are more probable in comparison to sea state' propagation direction parallel to the ship's movement direction. However, in case of higher incidence angle above 35° almost no influence of AIS-CoG-SAR-Wave-Direction is presented by the heatmaps. Higher incidence angles already imply better detectability of Kelvin wake arms and no more increase in detectability seems possible, when sea state and cusp waves are colliding.

The characteristics of influence of AIS-CoG-SAR-Wave-Direction on the detectability of V-narrow wakes are in contradiction between the port and starboard detectability models and the robustness checks also failed. Further, the influence is not pronounced. Therefore, no influence of AIS-CoG-SAR-Wave-Direction on the detectability of V-narrow wakes is assumed. The reason is that the Bragg scattering, which constitutes the V-narrow wakes,

should barely be affected be the direction of the interacting sea state. As described in the Sections 4.6 and 4.7, SAR-Significant-Wave-Height and SAR-Wave-Length are of importance.

*4.9. AIS-CoG-WRF-Wind-Direction*

The robustness check failed for turbulent wakes. Additionally, the characteristics of influences of AIS-CoG-WRF-Wind-Direction on the detectability of near-hull turbulences and turbulent wakes are inconsistent, when HH-polarization is compared to VV-polarization. The detectability model for Kelvin wake arms reproduces no influence for both polarizations. However, in [9] it is shown that the wind direction relative to the Kelvin wake arms has a similar influence for HH-polarization and VV-polarization. As these inconsistencies are in contradiction to some of the existing literature and additionally the characterization is not robust, it is assumed that the AIS-CoG-WRF-Wind-Direction parameter does not sufficiently represent the local relative wind direction and inconsistencies exist due to the noisy nature of this influencing parameter. In conclusion, it is assumed that AIS-CoG-WRF-Wind-Direction has no influence on the detectability of near-hull turbulences, turbulent wakes and Kelvin wake arms. As the resolution of state-of-the-art weather models cannot depict the local variability at the required scale, the only possibility would be to derive the wind direction from the SAR images or from local measurements, e.g., buoys. As the latter data source is only sparsely available, the investigation of the influence of the relative wind direction on the detectability of wake components is left for future research.

However, for V-narrow wakes the detectability models for port and starboard V-narrow wake arms robustly reproduce positive monotonic influence. Thus, when the wind strikes the wake signatures in a perpendicular direction to the ship's movement, the required first or second order Bragg wavelengths, which constitute the V-narrow wake arms, are promoted. According to [49,50], when winds are striking the smooth water region of turbulent wakes, the lee side of the turbulent wake gets rougher than the luv side, which results in a smooth line of bright backscatter being images by the SAR on the lee side. During training data generation for the present study those bright lines were only encountered, when ships were moving parallel to range direction. Here, those lines of brighter backscatter were interpreted as V-narrow wake arms. It should be noted that this interpretation might be incorrect, but anyway, V-narrow wakes are better detectable, when the ship movement is parallel to azimuth direction (Section 4.3).

**5. Conclusions**

In this work a large dataset of TerraSAR-X images, collocated in space and time with ground truth data from AIS, was applied to model the detectability of the components of ship wakes in dependency to parameters influencing the detectability. The individual wake components have been manually retraced in thousands of wake samples and the derived wake component's lengths are used as indicator for wake component detectability. A figure of merit, called Detectable Length Metric $DLM_w$, is introduced to measure the detectability. Machine learning, i.e., SVR, is used to model the $DLM_w$ of individual wake components in dependency to nine influencing parameters. These parameters include image acquisition settings (i.e., incidence angle), ship properties (i.e., ship's course of ground, ship's length and ship's velocity) and environmental conditions (i.e., wind speed, wind direction and sea state (significant wave height, wavelength and wave direction)). As an additional attribute, the polarization, under which the images are acquired, is considered in the SVR models. For the following wake components, detectability is analyzed by these means: near-hull turbulences, turbulent wakes, Kelvin wake arms and V-narrow wake arms. Transverse waves and divergent waves have an oscillating nature and therefore their detectability was only measured by the binary flags "detected" and "not detected", respectively, and their detectability was modelled using a binary SVM classifier similar to [12]. For ship-generated internal waves, insufficient data was available and therefore a model couldn't be created, but still some simpler qualitative statements are provided.

The results show that the characteristics of the influences of the parameters on the wake component detectability are mostly in agreement with statements and results provided by other researchers on the basis of SAR-image simulations or qualitative SAR image analysis together with theoretical considerations. Additionally, new results are provided which are presently missing in the literature. Finally, some results from the literature are contradicted.

The following list concludes the results and contradictions to previous results. New results are marked accordingly.

1. Near-hull turbulences (have not yet been considered separately from turbulent wakes)
   1.1 are better detectable, when the vessels move faster (new)
   1.2 are better detectable, when the vessels are larger (new)
   1.3 are better detectable, when the vessels move parallel to range direction (new, but not robust)
   1.4 are better detectable, when the incidence angles are larger (new)
   1.5 are better detectable, when the wind speeds are lower (new)
   1.6 are hardly influenced in detectability by the sea state's significant wave heights (new)
   1.7 are hardly influenced in detectability by the sea state's wavelengths (new)
   1.8 are hardly influenced in detectability by the sea state's wave directions (new)
   1.9 are hardly influenced in detectability by the WRF wind directions relative to the vessel's movement directions (new)

2. Turbulent wakes:
   2.1 are better detectable, when the vessels move faster (new)
   2.2 are better detectable, when the vessels are larger (new)
   2.3 are only under swell wave conditions and HH-polarization better detectable, when the vessels move parallel to azimuth direction. (conditions are new)
   2.4 are better detectable, when the incidence angles are smaller
   2.5 are better detectable, when the wind speeds are lower
   2.6 are hardly influenced in detectability by the sea state's significant wave heights (no agreement with previous research)
   2.7 are better detectable, when the sea state's wavelengths are longer (new)
   2.8 are hardly influenced in detectability by the sea state's wave directions (new)
   2.9 are hardly influenced in detectability by the WRF wind directions relative to the vessel's movement directions (no agreement with previous research)

3. Kelvin wake arms:
   3.1 are better detectable, when the vessels move faster
   3.2 are better detectable, when the vessels are larger
   3.3 are better detectable, when the vessels move parallel to azimuth direction
   3.4 are better detectable, when the incidence angles are smaller
   3.5 are better detectable, when the wind speeds are lower
   3.6 are hardly influenced in detectability by the sea state's significant wave heights (no agreement with previous research)
   3.7 are better detectable, when the sea state's wavelengths are longer (new)
   3.8 are better detectable, when the sea state's wave directions are perpendicular to the vessel's movement directions, but only for lower incidence angles (new)
   3.9 are hardly influenced in detectability by the WRF wind directions relative to the vessel's movement directions

4. Kelvin wake's divergent waves:
   4.1 are better detectable, when the vessels move faster
   4.2 are hardly influenced in detectability by the vessel's lengths (no agreement with previous research)
   4.3 are better detectable, when the vessels move neither parallel to range nor parallel to azimuth (new)

4.4     are better detectable, when the incidence angles are smaller (new)

4.5     are better detectable, when the wind speeds are lower (new)

5.    Kelvin wake's transverse waves:

5.1     are better detectable, when the vessels move faster (cut-off effect more relevant for detectability than wave amplitudes)

5.2     are better detectable, when the vessels are smaller (cut-off effect more relevant for detectability than wave amplitudes)

5.3     are better detectable, when the vessels move parallel to range direction (contradiction to Lyden et al. [8], who generally assumed positive independent monotonic influence)

5.4     are better detectable, when the incidence angles are smaller (new)

5.5     are hardly influenced in detectability by the wind speeds (new)

6.    V-narrow wake arms:

6.1     are better detectable, when the vessels move faster

6.2     are better detectable, when the vessels are larger (new)

6.3     are better detectable, when the vessels are moving parallel to azimuth

6.4     are better detectable, when the incidence angles are smaller

6.5     are better detectable, when the wind speeds are lower

6.6     are hardly influenced in detectability by the sea state's significant wave heights (no agreement with previous research)

6.7     are better detectable, when the sea state's wavelengths are longer (new)

6.8     are hardly influenced in detectability by the sea state's wave directions relative to the vessel's movement directions (new)

6.9     are better detectable, when the WRF wind direction is perpendicular to the ship's CoG (new)

7.    Ship-generated internal waves:

7.3     are better detectable, when the vessels are moving parallel to azimuth (new)

7.5     are better detectable, when the wind speeds are lower

Last but not least, V-narrow wakes are frequently detectable on X-Band SAR, i.e., on TS-X high resolution data. This is in contradiction to Lyden et al. [8], who proposed that V-narrow wakes would not be detectable by X-Band SARs. It should be noted that such studies were executed in the past, where the quality as well as quantity of sensor data was worse in comparison to today and additionally the available computing power was insufficient for ML applications. Perhaps the lower resolution of earlier SAR-missions led to this false conclusion, as many identified V-narrow wake cases on TS-X are very close to the turbulent wake's calm water region.

**Supplementary Materials:** The datasets used for training of the detectability models are available online at https://www.mdpi.com/2072-4292/13/2/165/s1.

**Funding:** This research received no external funding and the APC was funded by DLR.

**Institutional Review Board Statement:** Not applicable.

**Informed Consent Statement:** Not applicable.

**Data Availability Statement:** The datasets used for training of the detectability models have been uploaded as Supplementary Materials.

**Acknowledgments:** Special thanks goes to Andrey Pleskachevsky, Hendrik Rothe and James Imber for supporting their knowledge in the field of SAR oceanography, machine learning and statistics during discussions.

**Conflicts of Interest:** The authors declare no conflict of interest.

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
