# Peer review of "Non-Linear Modeling of Detectability of Ship Wake Components in Dependency to Influencing Parameters Using Spaceborne X-Band SAR"

_remotesensing, doi:10.3390/rs13020165_

Round 1

Reviewer 1 Report

This work proposed a wave detection method, a large dataset of TerraSAR-X images, collocated in space and time with ground truth data from AIS, was applied to model the detectability of the components of ship wakes in dependency to parameters influencing the detectability. This is an interesting work. There are some corrections that has to be carried out by the authors to enhance the quality of the paper. (1) Key word, remove “Support Vector Machine; ocean surface imaging” . (2) Page 2. Line 66-70, “It consists of two parts: … ” rewrite the sentence. You are combining two different sentences with a comma. (3) I advise the authors should list the contribution of your works in Section Introduction. (4) Page.12, Line 323. “SVMs have already proven beneficial in previous studies”. Which ones? The references are missing. (5) Authors should describe the meaning of figure in a paragraph in Sec.3.2. (6) Table 3 and 8, the format should to be uniform. (7) Convolution neural networks (CNNs) capable to achieve better performance for classification task. Have the authors considered using CNNs for classification tasks?

Author Response

Thank you, adressing of your comments can be found in the file: "Answer_Reviewer_1.docx"

Reviewer 2 Report

see attached file

Author Response

Thank you, adressing of your comments can be found in the file: "Answer_Reviewer_2".docx"
